# Tame a Wild Camera:
# In-the-Wild Monocular Camera Calibration

**Shengjie Zhu, Abhinav Kumar, Masa Hu, and Xiaoming Liu**
Department of Computer Science and Engineering,
Michigan State University, East Lansing, MI, 48824
{zhusheng, kumarab6, huynshen}@msu.edu, liuxm@cse.msu.edu

## Abstract

3D sensing for monocular in-the-wild images, *e.g.*, depth estimation and 3D object detection, has become increasingly important. However, the unknown intrinsic parameter hinders their development and deployment. Previous methods for the monocular camera calibration rely on specific 3D objects or strong geometry prior, such as using a checkerboard or imposing a Manhattan World assumption. This work instead calibrates intrinsic via exploiting the monocular 3D prior. Given an undistorted image as input, our method calibrates the complete 4 Degree-of-Freedom (DoF) intrinsic parameters. First, we show intrinsic is determined by the two well-studied monocular priors: monocular depthmap and surface normal map. However, this solution necessitates a low-bias and low-variance depth estimation. Alternatively, we introduce the incidence field, defined as the incidence rays between points in 3D space and pixels in the 2D imaging plane. We show that: 1) The incidence field is a pixel-wise parametrization of the intrinsic invariant to image cropping and resizing. 2) The incidence field is a learnable monocular 3D prior, determined pixel-wisely by up-to-sacle monocular depthmap and surface normal. With the estimated incidence field, a robust RANSAC algorithm recovers intrinsic. We show the effectiveness of our method through superior performance on synthetic and zero-shot testing datasets. Beyond calibration, we demonstrate downstream applications in image manipulation detection & restoration, uncalibrated two-view pose estimation, and 3D sensing. Codes/models are held here.

## 1 Introduction

Camera calibration is typically the first step in numerous vision and robotics applications [27, 46] that involve 3D sensing. Classic methods enable accurate calibration by imaging a specific 3D structure such as a checkerboard [49]. With the rapid growth of monocular 3D vision, there is an increasing focus on 3D sensing from in-the-wild images, such as monocular depth estimation, 3D object detection [35, 36], and 3D reconstruction [42, 43]. While 3D sensing techniques over in-the-wild images are developed, camera calibration for such in-the-wild images continues to pose significant challenges.

Classic methods for monocular calibration use strong geometry prior, such as using a checkerboard. However, such 3D structures are not always available in in-the-wild images. As a solution, alternative methods relax the assumptions. For example, [29] and [23] calibrate using common objects such as human faces and objects' 3D bounding boxes. Another significant line of research [7, 19, 34, 37, 51, 62, 66] is based on the Manhattan World assumption [15], which posits that all planes within a scene are either parallel or perpendicular to each other. This assumption is further relaxed [28, 30, 64] to estimate the lines that are either parallel or perpendicular to the direction of gravity. The intrinsic parameters are recovered by determining the intersected vanishing points of detected lines, assuming a central focal point and an identical focal length.

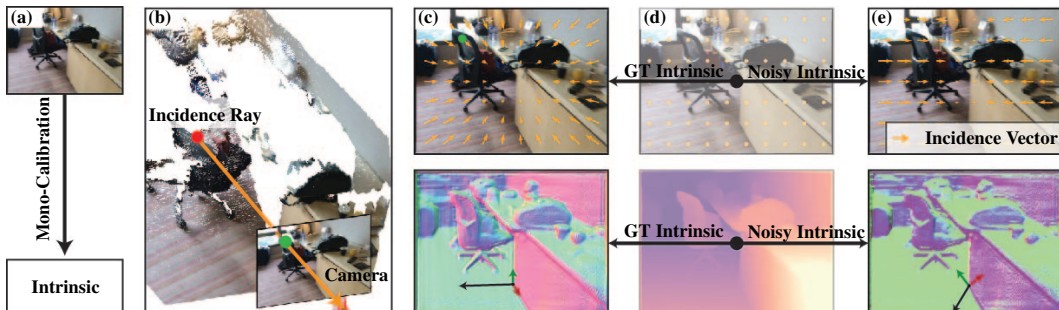

**Figure 1:** In (a), our work focuses on monocular camera calibration for in-the-wild images. We recover the intrinsic from monocular 3D-prior. In (c) - (e), an estimated depthmap is converted to surface normal using a groundtruth and noisy intrinsic individually. Noisy intrinsic distorts the point cloud, consequently leading to inaccurate surface normal. In (e), the normal presents a different color to (c). Motivated by the observation, we develop a solver that utilizes the **consistency** between the two to recover the intrinsic. However, this solution exhibits numerical instability. We then propose to learn the incidence field as an alternative 3D monocular prior. The incidence field is the collection of the pixel-wise incidence ray, which originates from a 3D point, targets at a 2D pixel, and crosses the camera origin, as shown in (b). Similar to depthmap and normal, a noisy intrinsic leads to a noisy incidence field, as in (e). By same motivation, we develop neural network to learn in-the-wild incidence field and develop a RANSAC algorithm to recover intrinsic from the estimated incidence field.

While the assumptions are relaxed, they may still not hold true for in-the-wild images. This creates a contradiction: although we develop robust models to estimate in-the-wild monocular depthmap, generating its 3D point cloud remains infeasible due to the missing intrinsic. A similar challenge arises in monocular 3D object detection, as we face limitations in projecting the detected 3D bounding boxes onto the 2D image. In AR/VR applications, the absence of intrinsic precludes placing multiple reconstructed 3D objects within a canonical 3D space. The absence of a reliable, assumption-free monocular intrinsic calibrator has become a bottleneck in deploying these 3D sensing applications.

Our method is motivated by the consistency between the monocular depthmap and surface normal map. In Fig. 1 (c) - (e), an incorrect intrinsic distorts the back-projected 3D point cloud from the depthmap, resulting in distorted surface normals. Based on this, intrinsic is optimal when the estimated monocular depthmap aligns consistently with the surface normal. We present a solution to recover the complete 4 DoF intrinsic by leveraging the consistency between the surface normal and depthmap. However, the algorithm is numerically ill-conditioned as its computation depends on the accurate gradient of depthmap. This requires depthmap estimation with low bias and variance.

To resolve it, we propose an alternative approach by introducing a novel 3D monocular prior in complementation to the depthmap and surface normal map. We refer to this as the incidence field, which depicts the incidence ray between the observed 3D point and the projected 2D pixel on the imaging plane, as shown in Fig. 1 (b). The combination of the incidence field and the monocular depthmap describes a 3D point cloud. Compared to the original solution, the incidence field is a direct pixel-wise parameterization of the camera intrinsic. This implies that a minimal solver based on the incidence field only needs to have low bias. We then utilize a deep neural network to perform the incidence field estimation. A non-learning RANSAC algorithm is developed to recover the intrinsic parameters from the estimated incidence field.

We consider the incidence field is also a monocular 3D prior. Similar to depthmap and surface normal, the incidence field is invariant to the image cropping or resizing. This encourages its generalization over in-the-wild images. Theoretically, we show the incidence field is pixel-wisely determined by depthmap and normal map. Empirically, to support our argument, we combine multiple public datasets into a comprehensive dataset with diverse indoor, outdoor, and object-centric images captured by different imaging devices. We further boost the variety of intrinsic by resizing and cropping the images in a similar manner as [23]. Finally, we include zero-shot testing samples to benchmark real-world monocular camera calibration performance.

We showcase downstream applications that benefit from monocular camera calibration. In addition to the aforementioned 3D sensing tasks, we present two intriguing additional applications. One is detecting and restoring image resizing and cropping. When an image is cropped or resized, it disrupts the assumption of a central focal point and identical focal length. Using the estimated

**Table 1: Camera Calibration Methods from Strong to Relaxed Assumptions.** Non-learning methods [23,29, 34,37,38,51,62,71–74] rely on strong assumptions. Learning-based methods [28,30,38] relax the assumptions to using gravity-aligned panorama images during training. Our method makes no assumptions except for undistorted images. This enables training with any calibrated images and calibrates 4 DoF intrinsic.

| | [23,29,38,71–74] | [34,37,51,62] | [28,38] | [30] | Ours |
|---|---|---|---|---|---|
| Degree of Freedom (DoF) | 4 | 1 | 1 | 3 | 4 |
| Assumption | Specific-Objects | Manhattan | Train with Panorama Images | | Undistorted Images |

intrinsic parameters, we restore the edited image by adjusting its intrinsic to a regularized form. The other application involves two-view uncalibrated camera pose estimation. With established image correspondence, a fundamental matrix [25] is determined. However, there does not exist an injective mapping between the fundamental matrix and camera pose [27]. This raises a counter-intuitive fact: inferring the pose from two uncalibrated images is infeasible. But our method enables uncalibrated two-view pose estimation by applying monocular camera calibration.

We summarize our contributions as follows:
⋄ Our approach tackles monocular camera calibration from a novel perspective by relying on monocular 3D priors. Our method makes no assumption for the to-be-calibrated undistorted image.
⋄ Our algorithm provides robust monocular intrinsic estimation for in-the-wild images, accompanied by extensive benchmarking and comparisons against other baselines.
⋄ We demonstrate its benefits on diverse and novel downstream applications.

## 2 Related Works

**Monocular Camera Calibration with Geometry.** One line of work [7,19,34,37,51,62,66] assumes the Manhattan World assumption [15], where all planes in 3D space are either parallel or perpendicular. Under the assumption, line segments in the image converge at the vanishing points, from which the intrinsic is recovered. LSD [61] and EDLine [2] develop robust line estimators. Others jointly estimate the horizon line and the vanishing points [40,54,70]. In Tab. 1, recent learning-based methods [28,30,38,63,64] relax the assumption by using the gravity-aligned panorama images during training. Such data provides the groundrtuth vanishing point and horizon lines which are originally computed with the Manhattan World assumption. Still, the assumption constrains [28,38,63,64] in modeling intrinsic as 1 DoF camera. Recently, [30] relaxes the assumption to 3 DoF via regressing the focal point. In comparison, our method makes no assumption except for undistorted images. This enables us to calibrate 4 DoF intrinsic and train with any calibrated images.

**Monocular Camera Calibration with Object.** Zhang's method [73] based on a checkerboard pattern is widely regarded as the standard for camera calibration. Several works generalize this method to other geometric patterns such as 1D objects [74], line segments [72], and spheres [71]. Recent works [29] and [23] extend camera calibration to real-world objects such as human faces. Optimizers, including BPnP [12] and PnP [56] are developed. However, the usage of specific objects restricts their applications. In contrast, our approach is applicable to any image.

**Image Cropping and Resizing.** Detecting photometric image manipulation [5,17,24] is extensively researched. But few studies detecting geometric manipulation, *e.g.*, resizing and cropping. On resizing, [18] regresses the image aspect ratio with a deep model. On cropping, a proactive method [67] is developed. We demonstrate that calibration also detects geometric manipulation. Our method does not need to encrypt images, complementing photometric manipulation detection.

**Uncalibrated Two-View Pose Estimation.** With the fundamental matrix estimated, the two-view camera pose is determined up to a projective ambiguity if images are uncalibrated. Alternative solutions [31,47,58] exist by employing deep networks to regress the pose. However, regression hinders the usage of geometric constraints, which proves crucial in calibrated two-view pose estimation [60,75,77]. Other works [20,26] use more than two uncalibrated images for pose estimation. Our work complements prior studies by enabling an uncalibrated two-view solution.

**Learnable Monocular 3D Priors.** Monocular depth [39,76] and surface normals [6] are two established 3D priors. Numerous studies have shown a learnable mapping between monocular images and corresponding 3D priors. Recently, [30] introduces the perspective field as a monocular 3D

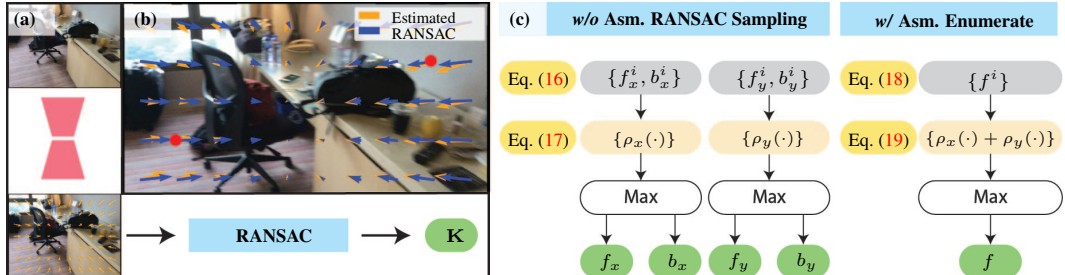

**Figure 2:** We illustrate the framework for the proposed monocular camera calibration algorithm. In (a), a deep network maps the input image $\mathbf{I}$ to the incidence field $\mathbf{V}$. A RANSAC algorithm recovers intrinsic from $\mathbf{V}$. In (b), we visualize a single iteration of RANSAC. An intrinsic is computed with two incidence vectors randomly sampled at red pixel locations. From Eq. (2), an intrinsic determines the incidence vector at a given location. The optimal intrinsic maximizes the consistency with the network prediction (blue and orange). Subfigure (c) details the RANSAC algorithm. Different strategies are applied depending on if a simple camera is assumed. If not assumed, we independently compute $(f_x, b_x)$ and $(f_y, b_y)$. If assumed, there is only 1 DoF of intrinsic. We proceed by enumerating the focal length within a predefined range to determine the optimal value.

prior for inferring gravity direction. In this work, we suggest the incidence field is also a learnable monocular 3D prior. A common characteristic of monocular 3D priors is their exclusive relationship with 3D structures, making them invariant to 2D manipulations, *e.g.*, cropping, and resizing. In contrast, camera intrinsic changes on spacial manipulation. Unlike other monocular priors, computing the groundtruth incidence field is straightforward, relying solely on intrinsic and image coordinates.

## 3 Method

Our work focuses on intrinsic estimation from in-the-wild monocular images captured by modern imaging devices. Hence we assume an undistorted image as input. In this section, we first show how to estimate intrinsic parameters by using monocular 3D priors, such as the surface normal map and monocular depthmap. We then introduce the incidence field and show it a learnable monocular 3D prior invariant to 2D spacial manipulation. Next, we describe the training strategy and the network used to learn the incidence field. After estimating the incidence field, we present a RANSAC algorithm to recover the 4 DoF intrinsic parameters. Lastly, we explore various downstream applications of the proposed algorithm. Fig. 2 shows the framework of the proposed algorithm.

### 3.1 Monocular Intrinsic Calibration

Our method aims to use generalizable monocular 3D priors without assuming the 3D scene geometry. Hence, we start with monocular depthmap $\mathbf{D}$ and surface normal map $\mathbf{N}$. Assume there exists a learnable mapping from the input image $\mathbf{I}$ to depthmap $\mathbf{D}$ and normal map $\mathbf{N}$: $\mathbf{D}, \mathbf{N} = \mathbb{D}_\theta(\mathbf{I})$, where $\mathbb{D}_\theta$ is a learned network. We denote the intrinsic $\mathbf{K}_{\text{simple}}$, $\mathbf{K}$, and its inverse $\mathbf{K}^{-1}$ as:

$$\mathbf{K}_{\text{simple}} = \begin{bmatrix} f & 0 & w/2 \\ 0 & f & h/2 \\ 0 & 0 & 1 \end{bmatrix}, \quad \mathbf{K} = \begin{bmatrix} f_x & 0 & b_x \\ 0 & f_y & b_y \\ 0 & 0 & 1 \end{bmatrix}, \quad \mathbf{K}^{-1} = \begin{bmatrix} 1/f_x & 0 & -b_x/f_x \\ 0 & 1/f_y & -b_y/f_y \\ 0 & 0 & 1 \end{bmatrix}. \quad (1)$$

The notation $\mathbf{K}_{\text{simple}}$ suggests a simple camera model with the identical focal length and central focal point assumption. Given a 2D homogeneous pixel location $\mathbf{p} = \begin{bmatrix} x & y & 1 \end{bmatrix}^{\mathsf{T}}$ and its depth value $d = \mathbf{D}(\mathbf{p})$, the corresponding 3D point $\mathbf{P} = \begin{bmatrix} X & Y & Z \end{bmatrix}^{\mathsf{T}}$ is defined as:

$$\mathbf{P} = \begin{bmatrix} X \\ Y \\ Z \end{bmatrix} = d \cdot \mathbf{K}^{-1} \begin{bmatrix} x \\ y \\ 1 \end{bmatrix} = d \cdot \begin{bmatrix} \frac{x - b_x}{f_x} \\ \frac{y - b_y}{f_y} \\ 1 \end{bmatrix} = d \cdot \begin{bmatrix} v_x \\ v_y \\ 1 \end{bmatrix} = d \cdot \mathbf{v}, \quad (2)$$

where the vector $\mathbf{v}$ is an incidence ray, originating from the 3D point $\mathbf{P}$, directed towards the 2D pixel $\mathbf{p}$, and passing through the camera's origin. The incidence field $\mathbf{V}$ is the collection of incidence rays $\mathbf{v}$ associated with all pixels at location $\mathbf{p}$, where $\mathbf{v} = \mathbf{V}(\mathbf{p})$.

### 3.2 Monocular Intrinsic Calibration with Monocular Depthmap and Surface Normal

In this section, we explain how to determine the intrinsic matrix $\mathbf{K}$ using the estimated surface normal map $\mathbf{N}$ and depthmap $\mathbf{D}$. Given the estimated depth $d = \mathbf{D}(\mathbf{p})$ and normal $\mathbf{n} = \mathbf{N}(\mathbf{p})$ at 2D pixel

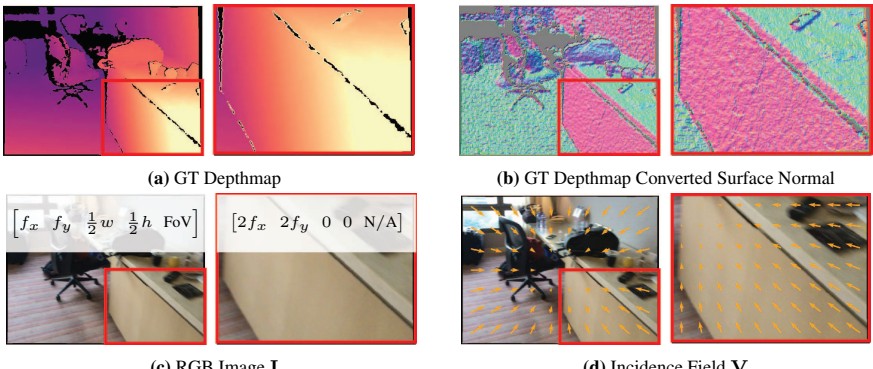

**(a)** GT Depthmap

**(b)** GT Depthmap Converted Surface Normal

**(c)** RGB Image $\mathbf{I}$

**(d)** Incidence Field $\mathbf{V}$

**Figure 3:** In (a) and (b), we highlight the ground truth depthmap of a smooth surface, such as a table's side. Even with the ground truth depthmap, the resulting surface normals exhibit noise patterns due to the inherent high variance. This makes the intrinsic solver based on the consistency of the depthmap and surface normals numerically unstable. Further, (a)-(d) demonstrate a scaling and cropping operation applied to each modality. In (c), the intrinsic changes per operation, leading to ambiguity if a network directly regresses the intrinsic values. Meanwhile, the FoV is undefined after cropping. In comparison, the incidence field remains invariant to image editing, the same as the surface normal and depthmap.

location $\mathbf{p}$, a local 3D plane is described as:

$$\mathbf{n}^\mathsf{T} \cdot d \cdot \mathbf{v} + c = 0. \tag{3}$$

By taking derivative in $x$-axis and $y$-axis directions, we have:

$$\mathbf{n}^\mathsf{T} \nabla_x (d \cdot \mathbf{v}) = 0, \quad \mathbf{n}^\mathsf{T} \nabla_y (d \cdot \mathbf{v}) = 0. \tag{4}$$

Note the bias $c$ of the 3D local plane is independent of the camera projection process. Without loss of generality, we show the case of our method for $x$-direction. Expanding Eq. (4), we obtain:

$$n_1 \nabla_x (d \cdot \frac{x - b_x}{f_x}) + n_2 \frac{y - b_y}{f_y} \nabla_x(d) + n_3 \nabla_x(d) = 0, \tag{5}$$

where $\nabla_x(d)$ represents the gradient of the depthmap $\mathbf{D}$ in the $x$-axis and can be computed, for example, using a Sobel filter [33]. Next, re-parametrize the unknowns in Eq. (5) to get:

$$a_1 f_x f_y + a_2 f_x b_y + a_3 f_y b_x + a_4 f_y + a_5 f_x = 0. \tag{6}$$

Divide both sides of the equation by $f_x$ to get:

$$a_1 f_y + a_2 b_y + a_3 r b_x + a_4 r + a_5 = 0, \tag{7}$$

where $r = \frac{f_y}{f_x}$. By stacking Eq. (7) with $N \geq 4$ randomly sampled pixels, we acquire a linear system:

$$\mathbf{A}_{N \times 4} \, \mathbf{X}_{4 \times 1} = \mathbf{B}_{4 \times 1}, \tag{8}$$

where the intrinsic parameter to be solved is stored in a vector $\mathbf{X}_{4 \times 1} = [f_y \quad b_y \quad r b_x \quad r]^\mathsf{T}$. This solves the other intrinsic parameters as:

$$f_y = f_y, \, b_y = b_y, \, f_x = \frac{f_y}{r}, \, b_x = \frac{r b_x}{r}. \tag{9}$$

The known constants are stored in matrix $\mathbf{A}_{N \times 4}$ and $\mathbf{B}_{4 \times 1}$. If we choose $N = 4$ in Eq. (8), we obtain a minimal solver where the solution $\mathbf{X}$ is computed by performing Gauss-Jordan Elimination. Conversely, when $N > 4$, the linear system is over-determined, and $\mathbf{X}$ is obtained using a least squares solver. The above suggests the intrinsic is recoverable from the monocular 3D prior.

### 3.3 Monocular Incidence Field as Monocular 3D Prior

Eq. (9) relies on the consistency between the surface normal and depthmap gradient, which may require a low-variance depthmap estimate. From Fig. 3, even groundtruth depthmap leads to spurious normal due to its inherent high variance. Thus, minimal solver in Eq. (9) can lead to a poor solution.

As a solution, we propose to directly learn the incidence field $\mathbf{V}$ as a monocular 3D prior. In Eq. (2) and Fig. 1, the combination of the incidence field $\mathbf{V}$ and the monocular depthmap $\mathbf{D}$ creates a 3D point cloud. In Eq. (3), the incidence field $\mathbf{V}$ can measure the observation angle between a 3D plane and the camera. Similar to depthmap $\mathbf{D}$ and surface normal map $\mathbf{N}$, the incidence field $\mathbf{V}$ is invariant to the image cropping and resizing. Consider an image cropping and resizing described as:

$$\mathbf{x}' = \Delta\mathbf{K}\,\mathbf{x}, \quad \Delta\mathbf{K} = \begin{bmatrix} \Delta f_x & 0 & \Delta c_x \\ 0 & \Delta f_y & \Delta c_y \\ 0 & 0 & 1 \end{bmatrix}, \quad \mathbf{K}' = \Delta\mathbf{K}\mathbf{K}, \tag{10}$$

where $\mathbf{K}'$ is the intrinsic after transformation. The surface normal map $\mathbf{N}$ and depthmap $\mathbf{D}$ after transformation is defined as:

$$\mathbf{N}'(\mathbf{x}') = \mathbf{N}(\mathbf{x}) = \mathbf{N}(\Delta\mathbf{K}^{-1}\mathbf{x}'), \quad \mathbf{D}'(\mathbf{x}') = \mathbf{D}(\mathbf{x}) = \mathbf{D}(\Delta\mathbf{K}^{-1}\mathbf{x}'). \tag{11}$$

Similarly, the incidence field after transformation is:

$$\mathbf{V}'(\mathbf{x}') = (\mathbf{K}')^{-1}\mathbf{x}' = \mathbf{K}^{-1}(\Delta\mathbf{K})^{-1}\,\mathbf{x}' = \mathbf{K}^{-1}\,\mathbf{x} = \mathbf{v} = \mathbf{V}(\mathbf{x}). \tag{12}$$

Eq. (12) shows that the incidence field $\mathbf{V}$ is a parameterization of the intrinsic matrix that is **invariant** to image resizing and image cropping. Other invariant parameterizations of the intrinsic matrix, such as the camera field of view (FoV), rely on the central focal point assumption and only cover a 2 DoF intrinsic matrix. An illustration is shown in Fig. 3.

Next, we prove the incidence field $\mathbf{V}$ is a monocular 3D prior pixel-wisely determined by monocular depth $\mathbf{D}$ and surface normal $\mathbf{N}$. Expanding Eq. (4) using the incidence vector $\mathbf{v}$ in Eq. (2), we get:

$$n_1 \nabla_x(d)v_x + n_1\left(\frac{d}{f_x}\right) + n_2 v_y \nabla_x(d) + n_3 \nabla_x(d) = 0. \tag{13}$$

Combined with the axis-$y$ constraint, the 2 DoF incidence vector $\mathbf{v} = \begin{bmatrix} v_x & v_y & 1 \end{bmatrix}^\mathsf{T}$ is uniquely solved. Eq. (13) gives identical solutions when depthmap $\mathbf{D}$ is adjusted by a scalar. This implies that the incidence field $\mathbf{V}$ is **pixel-wise determined** by the up-to-scale depth map and surface normal map, indicating a learnable mapping from the monocular image $\mathbf{I}$ to the incidence field $\mathbf{V}$ exists.

Given the strong connection between the monocular depthmap $\mathbf{D}$ and camera incidence field $\mathbf{V}$, we adopt NewCRFs [69], a neural network used in monocular depth estimation, for incidence field estimation. We change the last output head to output a three-dimensional normalized incidence field $\widetilde{\mathbf{V}}$ with the same resolution as the input image $\mathbf{I}$. We adopt a cosine similarity loss defined as:

$$\widetilde{\mathbf{V}} = \mathbb{D}_\theta(\mathbf{I}), \quad L = \frac{1}{N}\sum_{i=1}^{N}\widetilde{\mathbf{V}}^\mathsf{T}(\mathbf{x}_i)\widetilde{\mathbf{V}}_{\text{gt}}(\mathbf{x}_i). \tag{14}$$

We normalize the last dimension of the incidence field to one before feeding to the RANSAC algorithm. That is to say, $\mathbf{V}(\mathbf{x}_i) = \begin{bmatrix} \tilde{v}_1/\tilde{v}_3 & \tilde{v}_2/\tilde{v}_3 & 1 \end{bmatrix}^\mathsf{T} = \begin{bmatrix} v_1 & v_2 & 1 \end{bmatrix}^\mathsf{T}$.

### 3.4 Monocular Intrinsic Calibration with Incidence Field

Since the network inference executes on GPU device, we adopt a GPU-end RANSAC algorithm to recover the intrinsic $\mathbf{K}$ from the incidence field $\mathbf{V}$. Unlike a CPU-based RANSAC, we perform fixed $K_r$ iterations of RANSAC without termination. In RANSAC, we use the minimal solver to generate $K_c$ candidates and select the optimal one that maximizes a scoring function (see Fig. 2).

**RANSAC *w.o* Assumption.** From Eq. (2), the incidence vector $\mathbf{v}$ relates to the intrinsic $\mathbf{K}$ as:

$$\mathbf{v} = \mathbf{K}^{-1}\mathbf{x} = \begin{bmatrix} \frac{x-b_x}{f_x} & \frac{y-b_y}{f_y} & 1 \end{bmatrix}^\mathsf{T}. \tag{15}$$

From Eq. (15), a minimal solver for intrinsic is straightforward. In the incidence field, randomly sample two incidence vectors $\mathbf{v}^1 = \begin{bmatrix} v_x^1 & v_y^1 & 1 \end{bmatrix}^\mathsf{T}$ and $\mathbf{v}^2 = \begin{bmatrix} v_x^2 & v_y^2 & 1 \end{bmatrix}^\mathsf{T}$. The intrinsic is:

$$\begin{cases} f_x = \frac{x^1 - x^2}{v_x^1 - v_x^2} \\ b_x = \frac{1}{2}(x^1 - v_x^1 f_x + x^2 - v_x^2 f_x) \end{cases}, \quad \begin{cases} f_y = \frac{y^1 - y^2}{v_y^1 - v_y^2} \\ b_y = \frac{1}{2}(x^1 - v_y^1 f_y + x^2 - v_y^2 f_y) \end{cases}. \tag{16}$$

Similarly, the scoring function is defined in $x$-axis and $y$-axis, respectively:

$$\rho_x(f_x, b_x, \{\mathbf{x}\}, \{\mathbf{v}\}) = \sum_{i=1}^{N_k}\left(\left\|\frac{x^i - b_x}{f_x} - v_x^i\right\| < k_x\right), \quad \rho_y(f_y, b_y, \{\mathbf{x}\}, \{\mathbf{v}\}) = \sum_{i=1}^{N_k}\left(\left\|\frac{y^i - b_y}{f_y} - v_y^i\right\| < k_y\right), \tag{17}$$

**Table 2: In-the-Wild Monocular Camera Calibration.** We benchmark in-the-wild monocular camera calibration. We use the first 9 datasets for training, and test on all 14 datasets. Except for MegaDepth, we synthesize novel intrinsic by cropping and resizing during training. Note the synthesized images violate the focal point and focal length assumption. [**Key:** ZS = Zero-Shot, Asm. = Assumptions, Syn. = Synthesized]

| Dataset | Calibration | Scene | ZS | Syn. | Perspective [30] $e_f$ | $e_b$ | Ours $e_f$ | $e_b$ | Ours + Asm. $e_f$ | $e_b$ |
|---|---|---|---|---|---|---|---|---|---|---|
| NuScenes [10] | Calibrated | Driving | ✗ | ✔ | 0.378 | 0.286 | **0.102** | **0.087** | 0.402 | 0.400 |
| KITTI [22] | Calibrated | Driving | ✗ | ✔ | 0.631 | 0.279 | **0.111** | **0.078** | 0.383 | 0.368 |
| Cityscapes [14] | Calibrated | Driving | ✗ | ✔ | 0.624 | 0.316 | **0.108** | **0.110** | 0.387 | 0.367 |
| NYUv2 [53] | Calibrated | Indoor | ✗ | ✔ | 0.261 | 0.348 | **0.086** | **0.174** | 0.376 | 0.379 |
| ARKitScenes [8] | Calibrated | Indoor | ✗ | ✔ | 0.325 | 0.367 | **0.140** | **0.243** | 0.400 | 0.377 |
| SUN3D [65] | Calibrated | Indoor | ✗ | ✔ | 0.260 | 0.385 | **0.113** | **0.205** | 0.389 | 0.383 |
| MVImgNet [68] | SfM | Object | ✗ | ✔ | 0.838 | 0.272 | **0.101** | **0.081** | 0.108 | 0.072 |
| Objectron [1] | SfM | Object | ✗ | ✔ | 0.601 | 0.311 | **0.078** | **0.070** | 0.088 | 0.079 |
| MegaDepth [41] | SfM | Outdoor | ✗ | ✗ | 0.319 | **0.000** | 0.137 | 0.046 | **0.109** | **0.000** |
| Waymo [58] | Calibrated | Driving | ✔ | ✗ | 0.444 | **0.020** | 0.210 | 0.053 | **0.157** | **0.020** |
| RGBD [55] | Pre-defined | Indoor | ✔ | ✗ | 0.166 | **0.000** | 0.097 | 0.039 | **0.067** | **0.000** |
| ScanNet [16] | Calibrated | Indoor | ✔ | ✗ | 0.189 | **0.010** | 0.128 | 0.041 | **0.109** | **0.010** |
| MVS [21] | Pre-defined | Indoor | ✔ | ✗ | 0.185 | **0.000** | 0.170 | 0.028 | **0.127** | **0.000** |
| Scenes11 [11] | Pre-defined | Synthetic | ✔ | ✗ | 0.211 | **0.000** | 0.170 | 0.044 | **0.117** | **0.000** |

**Table 3: Comparisons to Monocular Camera Calibration with Geometry** on GSV dataset [4]. We follow the training and testing protocol of [38]. For a fair comparison, we convert the estimated intrinsic to camera FoV on the $y$-axis direction, following [30, 38], and report our results *w/* and *w/o* the assumptions.

| FoV (°) | Upright [37] PAMI'13 | Perceptual [28] CVPR'18 | CTRL-C [38] ICCV'21 | Perspective [30] CVPR'23 | Ours *w/o* Asm. | *w/* Asm. |
|---|---|---|---|---|---|---|
| Mean | 9.47 | 4.37 | 3.59 | 3.07 | 2.49 | **2.47** |
| Median | 4.42 | 3.58 | 2.72 | 2.33 | 1.96 | **1.92** |

where $N_k$ and $k_x/ k_y$ are the number of sampled pixels and the threshold for axis-$x/y$ directions.

**RANSAC *w/* Assumption.** If a simple camera model is assumed, *i.e.*, intrinsic only has an unknown focal length, it only needs to estimate 1-DoF intrinsic. We enumerate the focal length candidates as:

$$\{f\} = \{f_{\min} + \frac{i}{N_f}(f_{\max} - f_{\min}) \mid 0 \leq i \leq N_f\}. \tag{18}$$

The scoring function under the scenario is defined as the summation over $x$-axis and $y$-axis:

$$\rho(f, \{\mathbf{x}\}, \{\mathbf{v}\}) = \rho_x(f_x, w/2, \{\mathbf{x}\}, \{\mathbf{v}\}) + \rho_y(f_y, h/2, \{\mathbf{x}\}, \{\mathbf{v}\}). \tag{19}$$

### 3.5 Downstream Applications

**Image Crop & Resize Detection and Restoration.** Eq. (10) defines a crop and resize operation:

$$\mathbf{x}' = \Delta\mathbf{K}\,\mathbf{x}, \quad \mathbf{K}' = \Delta\mathbf{K}\,\mathbf{K}, \quad \mathbf{I}'(\mathbf{x}') = \mathbf{I}'(\Delta\mathbf{K}\,\mathbf{x}) = \mathbf{I}(\mathbf{x}). \tag{20}$$

When a modified image $\mathbf{I}'$ is presented, our algorithm calibrates its intrinsic $\mathbf{K}'$ and then:
Case 1: The original intrinsic $\mathbf{K}$ is known. *E.g.*, we obtain $\mathbf{K}$ from the image-associated EXIF file [3]. Image manipulation is computed as $\Delta\mathbf{K} = \mathbf{K}'\mathbf{K}^{-1}$. A manipulation is detected if $\Delta\mathbf{K}$ deviates from an identity matrix. The original image restores as $\mathbf{I}(\mathbf{x}) = \mathbf{I}'(\Delta\mathbf{K}\,\mathbf{x})$. Interestingly, the four corners of image $\mathbf{I}'$ are mapped to a bounding box in original image $\mathbf{I}$ under manipulation $\Delta\mathbf{K}$. We thus quantify the restoration by measuring the bounding box. See Fig. 4.
Case 2: The original intrinsic $\mathbf{K}$ is unknown. We assume the genuine image possess an identical focal length and central focal point. Any resizing and cropping are detected when matrix $\mathbf{K}'$ breaks this assumption. Note, the rule cannot detect aspect ratio preserving resize or centered crop. We restore the original image by defining an inverse operation $\Delta\mathbf{K}$ restore $\mathbf{K}'$ to an intrinsic fits the assumption.

**3D Sensing Related Tasks.** Intrinsic estimation can enable multiple applications. *E.g.*, depthmap to point cloud, uncalibrated two-view pose estimation, etc.

## 4 Experiments

**Implementation Details** Our network is trained using the Adam optimizer [32] with a batch size of 8. The learning rate is $1e^{-5}$, and the training process runs for $20,000$ steps. Our RANSAC algorithm

**Table 4: Comparisons to Monocular Camera Calibration with Object.** We compare to the recent Face-Calib [29], which calibrates the camera using video containing human faces. We report our results *w/* and *w/o* assuming a simple camera model. We perform **zero-shot** testing using the same model benchmarked in Tab. 2.

| Methods | BIWIRGBD-ID [48] | | | | | | CAD-120 [59] | | | | | |
|---|---|---|---|---|---|---|---|---|---|---|---|---|
| | $e_f$ | $e_{f_x}$ | $e_{f_y}$ | $e_b$ | $e_{b_x}$ | $e_{b_y}$ | $e_f$ | $e_{f_x}$ | $e_{f_y}$ | $e_b$ | $e_{b_x}$ | $e_{b_y}$ |
| Louraki [45] | 0.662 | 0.662 | 0.662 | - | 0.387 | 0.222 | 0.732 | 0.732 | 0.732 | - | 0.255 | 0.180 |
| Fetzer [20] WACV'20 | 0.845 | 0.845 | 0.845 | - | 0.001 | 0.005 | 0.679 | 0.679 | 0.679 | - | 0.001 | 0.005 |
| BPnP [12] CVPR'19 | 0.675 | 0.675 | 0.675 | - | 0.322 | 0.479 | 1.178 | 1.178 | 1.178 | - | 0.103 | 0.129 |
| FaceCalib [29] FG'23 | 0.133 | 0.133 | 0.133 | - | 0.026 | 0.042 | 0.151 | 0.151 | 0.151 | - | 0.023 | 0.063 |
| Ours | 0.034 | 0.029 | **0.016** | 0.020 | 0.011 | 0.018 | 0.137 | 0.137 | 0.054 | 0.042 | 0.042 | 0.008 |
| Ours + Assumptions | **0.019** | **0.019** | 0.019 | **0.000** | **0.000** | **0.000** | **0.047** | **0.047** | **0.047** | **0.000** | **0.000** | **0.000** |

is executed on a GPU, performing $K_r = 2,048$ iterations, with each iteration including a random sample of $K_c = 20,000$ incidence vectors. During both training and testing, the image is resized to a resolution of $480 \times 640$. The threshold for determining RANSAC inliers is $k_x = k_y = 0.02$.

**Evaluation Metrics** We follow [29] in assessing camera intrinsic estimation using the metrics:

$$e_{f_x} = \frac{\|f'_x - f_x\|}{f_x}, \quad e_{f_y} = \frac{\|f'_y - f_y\|}{f_y}, e_{b_x} = 2 \cdot \frac{\|b'_x - b_x\|}{w}, \quad e_{b_y} = 2 \cdot \frac{\|b'_y - b_y\|}{h}. \quad (21)$$

We additionally consolidate the performance along the x-axis and y-axis into comprehensive measurements, $e_f = \max(e_{f_x}, \ e_{f_y})$ and $e_b = \max(e_{b_x}, e_{b_y})$.

## 4.1 Monocular Camera Calibration In-The-Wild

**Datasets.** Our method is trained whenever a calibrated intrinsic is provided, making it applicable to a wide range of publicly available datasets. In Tab. 2, we incorporate datasets of different application scenarios, including indoor, outdoor scenes, driving, and object-centric scenes. Many of the datasets utilize only a single type of camera for data collection, resulting in a scarcity of intrinsic variations. Similar to [38], we employ random resizing and cropping to synthesize more intrinsic, marked in Tab. 2 column "Syn.". In augmentation, we first resize all images to a resolution of $480 \times 640$. We then uniformly random resize up to two times its size and subsequently crop to a resolution of $480 \times 640$. As MegaDepth [41] collects images captured by various cameras from the Internet, we disable its augmentation. We document the intrinsic parameters of each dataset in Supp.

In Tab. 2 column "Calibration", we assess intrinsic quality into various levels. "Calibrated" suggests accurate calibration with a checkerboard. "Pre-defined" is less accurate, indicating the default intrinsic provided by the camera manufacturer without a calibration process. "SfM" signifies that the intrinsic is computed via an SfM method [52].

**In-The-Wild Monocular Camera Calibration.** We benchmark in-the-wild monocular calibration performance on Tab. 2. For trained datasets, except for MegaDepth, we test on synthetic data using random cropping and resizing. For the unseen test dataset, we refrain from applying any augmentation to better mimic real-world application scenarios.

We compare to the recent baseline [30], which regresses intrinsic via a deep network. Note, [30] can not train on arbitrary calibrated images as requiring panorama images in training. A fair comparison using the same training and testing images is in Tab. 4 and Sec. 4.2. [30] provides models with two variations: one assumes a central focal point, and another does not. We report with the former model whenever the input image fits the assumption. From Tab. 2, our method demonstrates superior generalization across multiple unseen datasets. Further, the result *w/* assumption outperforms *w/o* assumption whenever the input images fit the assumption. Tested on an RTX-2080 Ti GPU, the combined network inference and calibration algorithm runs on average in 87 ms per image.

## 4.2 Monocular Camera Calibration with Geometry

Methods in this line of research hold a Manhattan World assumption, positing that images consist of planes that are either parallel or perpendicular to each other. Stated in Tab. 1, baselines [28, 30, 38] relax the assumption to training data. Our method imposes no other assumption except for undistorted images in both training and testing.

This brings three benefits. First, the assumption restricts their training to panorama images. In contrast, our model is trainable with any calibrated images. This yields improved generalization,

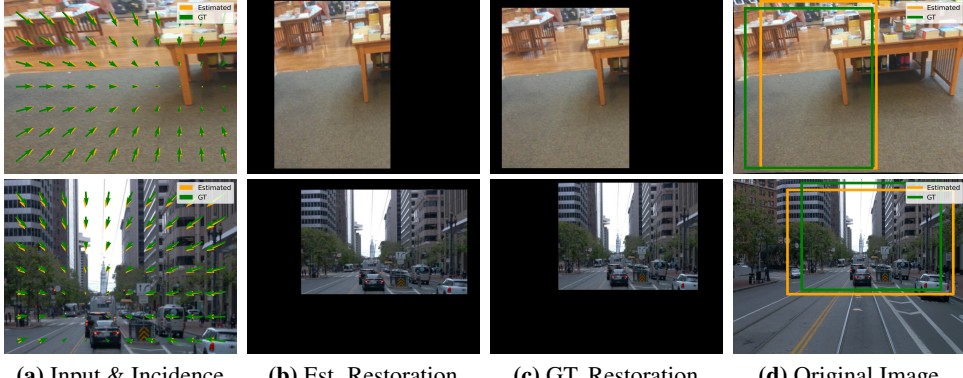

**(a)** Input & Incidence     **(b)** Est. Restoration     **(c)** GT. Restoration     **(d)** Original Image

**Figure 4: Image Crop & Resize Detection and Restoration.** Image editing, including cropping and resizing changes intrinsic. As in Sec. 3.5, monocular calibration is applicable to detect and restore image manipulations. We visualize the zero-shot samples on ScanNet and Waymo. More examples are in Supp.

**Table 5: Image Crop and Resize Restoration.** Stated in Sec. 3.5, our method also encompasses the restoration of image manipulations. Use model reported in Tab. 2, we conduct evaluations on both seen and unseen datasets.

| Methods | KITTI [22] | | NYUv2 [53] | | ARKitScenes [8] | | Waymo [58] | | RGBD [55] | | ScanNet [16] | | MVS [21] | |
|---|---|---|---|---|---|---|---|---|---|---|---|---|---|---|
| | mIOU | Acc | mIOU | Acc | mIOU | Acc | mIOU | Acc | mIOU | Acc | mIOU | Acc | mIOU | Acc |
| Baseline | 0.686 | 0.795 | 0.621 | 0.710 | 0.586 | 0.519 | 0.581 | 0.721 | 0.636 | 0.681 | 0.597 | 0.811 | 0.595 | 0.667 |
| Ours | **0.842** | **0.852** | **0.779** | **0.856** | **0.691** | **0.837** | **0.681** | **0.796** | **0.693** | **0.781** | **0.709** | **0.887** | **0.638** | **0.795** |

as shown in Tab. 2. Second, it constrains the baselines to a simple camera parameterized by FoV. We consider the proposed incidence field a more generalizable and invariant parameterization of intrinsic. *E.g.*, while FoV remains invariant to image resizing, it still changes after cropping. However, the incidence field is unaffected in both cases. In Tab. 3, the substantial improvement we achieved $(0.60 = 3.07 - 2.47)$ over the recent SoTA [30] empirically supports our argument. Third, our method calibrates the 4 DoF intrinsic with a non-learning RANSAC algorithm. Baselines instead regress the intrinsic. This renders our method more robust and interpretable. In Fig. 2 (b), the estimated intrinsic quality is visually discerned through the consistency achieved between the two incidence fields.

### 4.3 Monocular Camera Calibration with Objects

We compare to the recent object-based camera calibration method FaceCalib [29]. The baseline employs a face alignment model to calibrate the intrinsic over a video. Both [29] and ours perform zero-shot prediction. We report performance using Tab. 2 model. Compared to [29], our method is more general as it does not assume a human face present in the image. Meanwhile, [29] calibrates over a **video**, while ours is a **monocular** method. For a fair comparison, we report the video-based results as an averaged error over the videos. We report results *w/* and *w/o* assuming a simple camera model. Since the tested image has a central focal point, when the assumption applied, the error of the focal point diminished. In Tab. 4, we outperform SoTA substantially.

### 4.4 Downstream Applications

**Image Crop & Resize Detection and Restoration.** In Sec. 3.5, our method also addresses the detection and restoration of geometric manipulations in images. Using the model reported in Tab. 2, we benchmark its performance in Tab. 5. Random manipulations following Sec. 4.1 contribute to 50% of both train and test sets, and the other 50% are genuine images. In Tab. 5, we evaluate restoration with mIOU and report detection accuracy (*i.e.* binary classification of genuine vs edited images). From the table, our method generalizes to the unseen dataset, achieving an averaged mIOU of 0.680. Meanwhile, we substantially outperform the baseline, which is trained to directly regresses the intrinsic. The improvement suggests the benefit of the incidence field as an invariant intrinsic parameterization. Beyond performance, our algorithm is interpretable. In Fig. 4, the perceived image geometry is interpretable for humans.

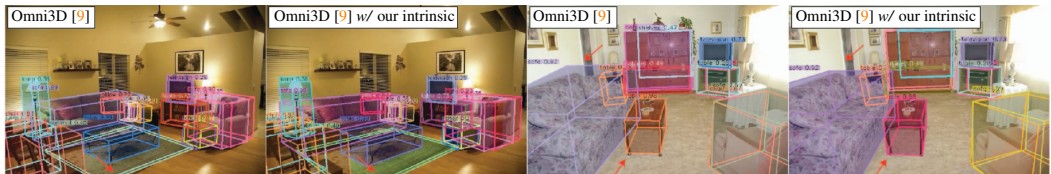

**Figure 5: In-the-Wild Camera Calibration Benefits In-the-Wild 3D Object Detection [9].** Our estimated intrinsic improves 2D projection of estimated 3D bounding boxes [9], highlighted with red arrows.

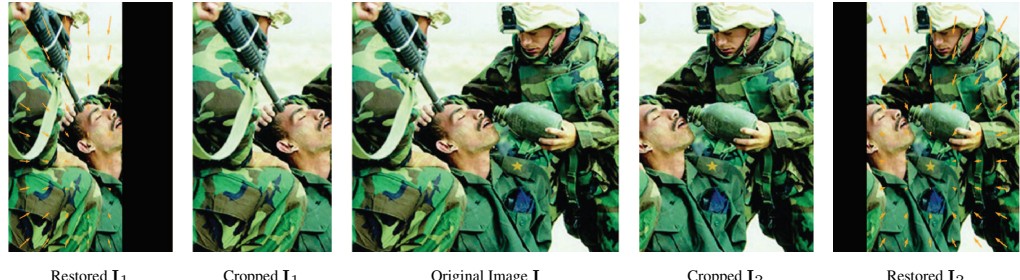

| Restored $\mathbf{I}_1$ | Cropped $\mathbf{I}_1$ | Original Image $\mathbf{I}$ | Cropped $\mathbf{I}_2$ | Restored $\mathbf{I}_2$ |

**Figure 6: Broader Impact.** Our method detects geometric manipulation and helps press image authenticity. In the figure, our method restores cropping and the aspect ratio. Orange arrows mark estimated incidence field.

**Table 6: Uncalibrated Two-View Camera Pose Estimation.** We use the model reported in Tab. 2 and assume distinct camera models for **both** frames. During calibration, we apply the simple camera assumption. The last two rows ablate the performance using GT intrinsic and our estimated intrinsic.

| Methods | Calibrated | ScanNet [16] | | | MegaDepth [41] | | |
|---|---|---|---|---|---|---|---|
| | | @5° | @10° | @20° | @5° | @10° | @20° |
| SuperGlue [50] CVPR'19 | ✔ | 16.2 | 33.8 | 51.8 | 42.2 | 61.2 | 75.9 |
| DRC-Net [44] ICASSP'22 | ✔ | 7.7 | 17.9 | 30.5 | 27.0 | 42.9 | 58.3 |
| LoFTR [57] CVPR'21 | ✔ | 22.0 | 40.8 | 57.6 | 52.8 | 69.2 | 81.2 |
| ASpanFormer [13] ECCV'22 | ✔ | 25.6 | 46.0 | 63.3 | 55.3 | 71.5 | 83.1 |
| PMatch [78] CVPR'23 | ✔ | 29.4 | 50.1 | 67.4 | 61.4 | 75.7 | 85.7 |
| PMatch [78] CVPR'23 | ✗ | 11.4 | 29.8 | 49.4 | 16.8 | 30.6 | 47.4 |

**Uncalibrated Two-View Camera Pose Estimation.** With correspondence between two images, one can infer the fundamental matrix [25]. However, the pose between two uncalibrated images is determined by a projective ambiguity. Our method eliminates the ambiguity with monocular camera calibration. In Tab. 6, we benchmark the uncalibrated two-view pose estimation and compare it to recent baselines. The result is reported using the model benchmarked in Tab. 2 and assumes unique intrinsic for **both** images. We perform zero-shot in ScanNet. For MegaDepth, it includes images collected over the Internet with diverse intrinsics. Interestingly, in ScanNet, our uncalibrated method outperforms a calibrated one [44]. In Supp, we plot the curve between pose performance and intrinsic quality. The challenging setting suggests itself an ideal task to evaluate the intrinsic quality.

**In-the-Wild Monocular 3D Sensing.** In-the-wild 3D sensing, *e.g.*, monocular 3D object detection [9], requires intrinsic to connect the 3D and 2D space. In Fig. 5, despite 3D bounding boxes being accurately estimated, incorrect intrinsic produces suboptimal 2D projection. Our calibration results suggest monocular incidence field estimation is another essential monocular 3D sensing task.

## 5 Conclusion

We calibrate monocular images through a novel monocular 3D prior referred as incidence field. The incidence field is a pixel-wise parameterization of intrinsic invariant to image resizing and cropping. A RANSAC algorithm is developed to recover intrinsic from the incidence field. We extensively benchmark our algorithm and demonstrate robust in-the-wild performance. Beyond calibration, we show multiple downstream applications that benefit from our method.

**Limitation.** In real application, whether to apply the assumption still requires human input.

**Broader Impacts.** Our method detects image spacial editing. If the Exif files are provided, it further localizes the modification region. Shown in Fig. 6, it helps improve press image authenticity.

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
