# Tame a Wild Camera:
# In-the-Wild Monocular Camera Calibration
# ======= Supplementary =======

**Shengjie Zhu, Abhinav Kumar, Masa Hu, and Xiaoming Liu**
Department of Computer Science and Engineering,
Michigan State University, East Lansing, MI, 48824
{zhusheng, kumarab6, huynshen}@msu.edu, liuxm@cse.msu.edu

## 1   Restore Cropping and Resizing without the Original Intrinsic

In the main paper Sec. 3.6, when the original intrinsic is unknown, we restore the modified image by defining an inverse operation $\Delta\mathbf{K}$ to restore $\mathbf{K}'$ to an intrinsic follow the simple camera assumption. Suppose the input image $\mathbf{I}'$ of size $(w \times h)$, we apply monocular camera calibration to estimate its intrinsic as $\mathbf{K}'$. We introduce the reverse operation as:

$$\mathbf{K}' = \begin{bmatrix} f'_x & 0 & b'_x \\ 0 & f'_y & b'_y \\ 0 & 0 & 1 \end{bmatrix}, \ \Delta\mathbf{K}_f = \begin{bmatrix} 1 & 0 & 0 \\ 0 & f'_x/f'_y & 0 \\ 0 & 0 & 1 \end{bmatrix}, \ \Delta\mathbf{K}_b = \begin{bmatrix} 1 & 0 & \Delta b_x \\ 0 & 1 & \Delta b_y \\ 0 & 0 & 1 \end{bmatrix}. \tag{1}$$

Set $r' = f'_x/f'_y$, the $\Delta b'_x$ and $\Delta b'_y$ are conditioned as:

$$\Delta b_x = \begin{cases} 0, & \text{if } b'_x \geq w - b'_x \\ w - 2b'_x, & \text{otherwise} \end{cases}, \quad \Delta b_y = \begin{cases} 0, & \text{if } b'_y \geq h - b'_y \\ r' \cdot (h - 2b'_y), & \text{otherwise} \end{cases}. \tag{2}$$

The restoration operation is defined as $\Delta\mathbf{K} = \Delta\mathbf{K}_b\Delta\mathbf{K}_f$. After the restoration, the width of the new image is $w' = 2\max(w - b'_x, \ b'_x)$ and the height of the new image is $h' = 2r'\max(h - b'_y, \ b'_y)$. An illustration depicting the restoration process is presented in Fig. 1.

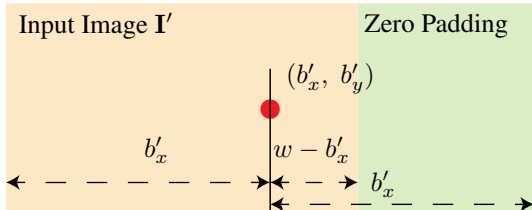

**Figure 1:** We illustrates the construction of $\Delta\mathbf{K}_b$ when $b'_x \geq w - b'_x$. We mark the focal point using a red dot. To position the focal point at the center of the image, we pad the right side of the images with zeros. When padding is applied to the right side of the image, the origin of the 2D image coordinate system remains unchanged. Therefore, we set $b'_x$ to be 0. Conversely, when padding is applied to the left of the image, we must assign a non-zero value to $b'_x$ in order to account for the shift in image coordinates.

## 2   Experiments

### 2.1   In-the-Wild Monocular Camera Calibration

**Train, Validation, and Test Split.**   We randomly sample 800 images from the training set to formulate the validation split. For SUN3D, MVS, Scenes11, and RGBD, we follow [23]. For

37th Conference on Neural Information Processing Systems (NeurIPS 2023).

**Table 1: Dataset Statistics.** We document the intrinsic, focal point, and camera FoV (in degree) in the table. The upper and lower part suggests seen and unseen datasets during training. [**Key:** Syn. = Synthesized]

| Dataset | Calibration | Scene | Syn. | $f_x$ | $f_y$ | $b_x$ | $b_y$ | $w$ | $h$ | FoV$_x$ | FoV$_y$ |
|---|---|---|---|---|---|---|---|---|---|---|---|
| NuScenes [5] | Calibrated | Driving | ✔ | 1266.42 | 1266.42 | 816.27 | 491.51 | 1600 | 900 | 64.56 | 39.12 |
| KITTI [11] | Calibrated | Driving | ✔ | 718.86 | 718.86 | 607.19 | 185.22 | 1241 | 376 | 81.60 | 29.31 |
| Cityscapes [7] | Calibrated | Driving | ✔ | 2267.86 | 2230.28 | 1045.53 | 518.88 | 2048 | 1024 | 48.60 | 25.86 |
| NYUv2 [17] | Calibrated | Indoor | ✔ | 518.85 | 519.47 | 325.58 | 253.74 | 640 | 480 | 63.33 | 49.59 |
| SUN3D [24] | Calibrated | Indoor | ✔ | 570.34 | 570.32 | 320.00 | 240.00 | 640 | 480 | 58.59 | 45.64 |
| ARKitScenes [3] | Calibrated | Indoor | ✔ | 1601.95 | 1601.95 | 936.55 | 709.61 | 1920 | 1440 | 62.15 | 48.65 |
| Objectron [1] | Calibrated | Object | ✔ | 1579.18 | 1579.18 | 721.01 | 934.70 | 1440 | 1920 | 48.53 | 62.01 |
| MVImgNet [25] | SfM | Object | ✔ | | | | Varying Intrinsic | | | | |
| MegaDepth [16] | SfM | Outdoor | ✗ | | | | Varying Intrinsic | | | | |
| Waymo [20] | Calibrated | Driving | ✗ | 2060.56 | 2060.56 | 947.46 | 634.37 | 1920 | 1280 | 49.73 | 34.34 |
| RGBD [18] | Pre-defined | Indoor | ✗ | 570.00 | 570.00 | 320.00 | 240.00 | 640 | 480 | 58.62 | 45.67 |
| ScanNet [8] | Calibrated | Indoor | ✗ | 1165.72 | 1165.74 | 649.09 | 484.77 | 1296 | 968 | 58.30 | 45.33 |
| MVS [10] | Pre-defined | Hybrid | ✗ | 570.34 | 570.34 | 320.00 | 240.00 | 640 | 480 | 58.59 | 45.64 |
| Scenes11 [6] | Pre-defined | Synthetic | ✗ | 570.00 | 570.00 | 320.00 | 240.00 | 640 | 480 | 58.62 | 45.67 |

ScanNet and MegaDepth, we follow [27]. For ARKitScenes and Objectron, we follow [4]. For NuScenes, CityScapes, and Waymo, we follow the official train split and use the validation as the test split. For KITTI, we use the sequences collected in date "2011_10_03" as testing split and others as training split. For MVImgNet, we use the "MVImgNet_42.zip" for testing and the first 4 zip files for training. For NYUv2, we follow [22]. To address the excessive image counts in certain test splits, we randomly downsample each dataset's test set to 800 images. Note, since SUN3D, MVS, Scenes11, and RGBD only contain 160 images, we include all of them as the testing set.

**Intrinsic Documentation.** In Tab. 1, we record the intrinsic of training and testing data without augmentation. Many datasets in Tab. 1, such as the KITTI dataset, conduct calibration multiple times, leading to slight variations in their intrinsic. Since the difference is minor, we opt to record the intrinsic of the first test sample for each dataset.

MegaDepth gathers images captured by diverse imaging devices from the internet, resulting in a wide range of intrinsic. We denote this specific scenario with the term "Varying Intrinsic". For MVImgNet, while it is generated using SfM similar to MegaDepth, its images are collected with a single type of camera. Therefore, while we document its intrinsic as "Varying Intrinsic", we still apply augmentation. In MVImgNet and Objectron datasets, which emphasize object-centric images, augmentations have a tendency to remove foreground objects, leaving behind textureless backgrounds. To prevent this, we reduce the augmentation by 1/5 compared to other synthetic datasets.

**Training and Testing Intrinsic Distribution.** In Fig. 2, we plot the training and testing set intrinsic distribution of the seen datasets, *i.e.*, the upper half of Tab. 1. We use camera FoV as a normalized measurement to indicate intrinsic variations. Since we include image cropping during synthesis, we introduce a generalized camera FoV for cropped images, defined as:

$$\text{FoV}_x = \arctan(\frac{w - b_x}{f_x}) - \arctan(\frac{0 - b_x}{f_x}), \quad \text{FoV}_y = \arctan(\frac{h - b_y}{f_y}) - \arctan(\frac{0 - b_y}{f_y}). \quad (3)$$

We report the FoV distributions in Fig. 2. From Fig. 2, MegaDepth (marked in brown dots) gathers images captured by diverse imaging devices from the Internet, resulting in a wide range of intrinsic. The large variation of MegaDepth intrinsic makes it an ideal dataset for benchmarking monocular camera calibration. In the main paper Tab. 2, our method achieves superior performance on MegaDepth dataset. This further validates the generalization capability of our method.

In Fig. 2, we also compare our experimental setting with our baselines [12–15]. Our baselines set up the experiment (main paper Tab. 3) on a single dataset GSV using synthesized images with FoV ranging from $40°$ to $80°$. Since they assume identical FoV for image axis X and Y directions, their FoV distribution is represented by a Red Line. From Fig. 2, our experiments include images collected from multiple datasets with diverse augmentation. This makes our experiments far more challenging and comprehensive compared to the baselines.

**In-the-Wild Monocular Calibration without SUN3D dataset.** In Tab. 1, for indoor datasets SUN3D, RGBD, Scenes11, and hybrid datasets MVS, they share a similar intrinsic. But each of them is captured with different devices. SUN3D is collected with SUSXtion PRO LIVE [21]. RGBD uses Microsoft Kinect [26]. Scenes11 uses synthetically generated images [23]. The MVS dataset

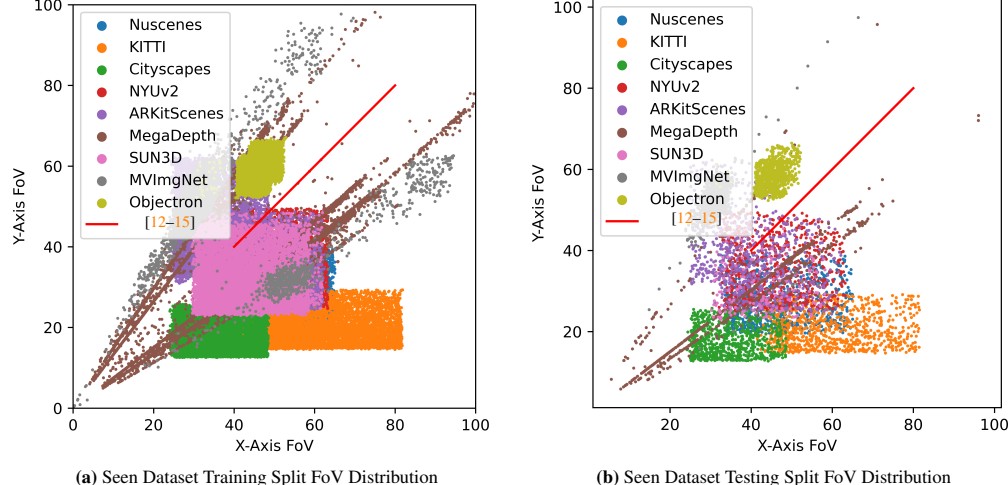

**(a)** Seen Dataset Training Split FoV Distribution      **(b)** Seen Dataset Testing Split FoV Distribution

**Figure 2:** We plot the Camera FoV distribution of the seen dataset training and testing split. We compare it to the experimental setting with our baseline works [12–15]. The baselines synthesize images with the FoV ranging between $40°$ to $80°$ on the GSV dataset [2] only (main paper Tab. 3). Since [12–15] assume identical camera FoV on image axis X and Y direction, their FoV distribution is represented by the single Red Line. In contrast, we adopt a significantly more challenging and comprehensive experimental setting. We synthesize images over multiple datasets without an assumption of identical camera FoV. Further, we include cropping in the synthesis. Combining both points, our training and testing data possess a diverse FoV distribution. Additionally, a small FoV is in general more challenging for calibration since the distortion from camera projection is minor at a small FoV. Our experiments contain diverse small FoVs compared to baselines. During plotting, we compute a generalized camera FoV if the image is cropped defined in Eq. (3).

**Table 2: In-the-Wild Monocular Camera Calibration Excludes SUN3D Dataset.** We repeat the main paper Tab. 2 experiment after excluding the SUN3D dataset in training. The exclusion of SUN3D dataset does not change the conclusion. We use ↓ and ↑ to indicate an improved and inferior performance compared to the main paper Tab. 2. [**Key:** ZS = Zero-Shot, Asm. = Assumptions, Syn. = Synthesized]

| Dataset | Calibration | Scene | ZS | Syn. | Perspective [13] $e_f$ | Perspective [13] $e_b$ | Ours $e_f$ | Ours $e_b$ | Ours + Asm. $e_f$ | Ours + Asm. $e_b$ |
|---|---|---|---|---|---|---|---|---|---|---|
| NuScenes [5] | Calibrated | Driving | ✗ | ✔ | 0.610 | 0.248 | **0.066**↓ | **0.082** | 0.372 | 0.400 |
| KITTI [11] | Calibrated | Driving | ✗ | ✔ | 0.670 | 0.221 | **0.081**↓ | **0.113** | 0.420 | 0.368 |
| Cityscapes [7] | Calibrated | Driving | ✗ | ✔ | 0.713 | 0.334 | **0.074**↓ | **0.081** | 0.383 | 0.367 |
| NYUv2 [17] | Calibrated | Indoor | ✗ | ✔ | 0.449 | 0.409 | **0.076**↓ | **0.163** | 0.332 | 0.379 |
| ARKitScenes [3] | Calibrated | Indoor | ✗ | ✔ | 0.362 | 0.410 | **0.085**↓ | **0.164** | 0.338 | 0.377 |
| MVImgNet [25] | SfM | Object | ✗ | ✔ | 0.204 | 0.500 | **0.074**↓ | **0.065** | 0.085 | 0.072 |
| Objectron [1] | Calibrated | Object | ✗ | ✔ | 0.178 | 0.339 | **0.054**↓ | **0.069** | 0.063 | 0.079 |
| MegaDepth [16] | SfM | Outdoor | ✗ | ✗ | 0.493 | **0.000** | 0.138 | 0.056 | **0.112**↑ | **0.000** |
| SUN3D [24] | Calibrated | Indoor | ✔ | ✗ | 0.260 | 0.271 | 0.094 | 0.051 | **0.086** | **0.000** |
| Waymo [20] | Calibrated | Driving | ✔ | ✗ | 0.564 | **0.020** | 0.199 | 0.030 | **0.150**↓ | **0.020** |
| RGBD [18] | Pre-defined | Indoor | ✔ | ✗ | 0.264 | **0.000** | 0.136 | 0.058 | **0.103**↑ | **0.000** |
| ScanNet [8] | Calibrated | Indoor | ✔ | ✗ | 0.385 | **0.010** | 0.169 | 0.020 | **0.150**↑ | **0.010** |
| MVS [10] | Pre-defined | Indoor | ✔ | ✗ | 0.312 | **0.000** | 0.198 | 0.027 | **0.130**↑ | **0.000** |
| Scenes11 [6] | Pre-defined | Synthetic | ✔ | ✗ | 0.348 | **0.000** | 0.196 | 0.038 | **0.157**↑ | **0.000** |

comprises a combination of default downsampled high-resolution images from COLMAP and images captured using cellphones [23]. ScanNet shares a similar camera FoV. But ScanNet is collected with an iPad camera [8]. The problem arises as we include the SUN3D dataset in our training set. However, since one of the objectives of this research is to provide a beneficial model for other researchers, it is reasonable to incorporate a widely adopted intrinsic pattern. Despite it, people may still question whether the inclusion of SUN3D decisively influences the zero-shot performance of unseen datasets RGBD, Scenes11, MVS, and ScanNet, since the intrinsic has been seen during training. We answer this question via re-experimenting the main paper Tab. 2 after excluding the SUN3D.

We report the results in Tab. 2. From Tab. 2, the exclusion of SUN3D leads to improved performance on Waymo and inferior performance on RGBD, Scenes11, MVS, and ScanNet datasets. This suggests the inclusion of SUN3D helps generalize datasets with similar intrinsic values. One interesting discovery is that the removal of SUN3D consistently results in improvements across synthetic datasets. We interpret this as an indication that SUN3D exhibits a distinct distribution in comparison

**Table 3: In-the-Wild Monocular Camera Calibration with Augmetnation.** We synthesize novel zero-shot intrinsic with augmentation following main paper Sec. 4.1 to the **unseen** dataset. We add new results to the last 5 rows of the table. The rest of the table is identical to the main paper Tab. 2. Note, synthesis breaks the simple camera assumption. Fig. 4 to Fig. 8 visualize the intrinsic estimation of the last 5 rows via applying same augmentation. [**Key:** ZS = Zero-Shot, Asm. = Assumptions, Syn. = Synthesized]

| Dataset | Calibration | Scene | ZS | Syn. | Perspective [13] $e_f$ | $e_b$ | Ours $e_f$ | $e_b$ | Ours + Asm. $e_f$ | $e_b$ |
|---|---|---|---|---|---|---|---|---|---|---|
| NuScenes [5] | Calibrated | Driving | ✗ | ✔ | 0.610 | 0.248 | **0.102** | **0.087** | 0.402 | 0.400 |
| KITTI [11] | Calibrated | Driving | ✗ | ✔ | 0.670 | 0.221 | **0.111** | **0.078** | 0.383 | 0.368 |
| Cityscapes [7] | Calibrated | Driving | ✗ | ✔ | 0.713 | 0.334 | **0.108** | **0.110** | 0.387 | 0.367 |
| NYUv2 [17] | Calibrated | Indoor | ✗ | ✔ | 0.449 | 0.409 | **0.086** | **0.174** | 0.376 | 0.379 |
| ARKitScenes [3] | Calibrated | Indoor | ✗ | ✔ | 0.362 | 0.410 | **0.140** | **0.243** | 0.400 | 0.377 |
| SUN3D [24] | Calibrated | Indoor | ✗ | ✔ | 0.442 | 0.501 | **0.113** | **0.205** | 0.389 | 0.383 |
| MVImgNet [25] | SfM | Object | ✗ | ✔ | 0.204 | 0.500 | **0.101** | **0.081** | 0.108 | 0.072 |
| Objectron [1] | Label | Object | ✗ | ✔ | 0.178 | 0.339 | **0.078** | **0.070** | 0.088 | 0.079 |
| MegaDepth [16] | SfM | Outdoor | ✗ | ✗ | 0.493 | **0.000** | 0.137 | 0.046 | **0.109** | **0.000** |
| Waymo [20] | Calibrated | Driving | ✔ | ✗ | 0.564 | **0.020** | 0.210 | 0.053 | **0.157** | **0.020** |
| RGBD [18] | Pre-defined | Indoor | ✔ | ✗ | 0.264 | **0.000** | 0.097 | 0.039 | **0.067** | **0.000** |
| ScanNet [8] | Calibrated | Indoor | ✔ | ✗ | 0.385 | **0.010** | 0.128 | 0.041 | **0.109** | **0.010** |
| MVS [10] | Pre-defined | Indoor | ✔ | ✗ | 0.312 | **0.000** | 0.170 | 0.028 | **0.127** | **0.000** |
| Scenes11 [6] | Pre-defined | Synthetic | ✔ | ✗ | 0.348 | **0.000** | 0.170 | 0.044 | **0.117** | **0.000** |
| Waymo [20] | Calibrated | Driving | ✔ | ✔ | 0.655 | 0.266 | **0.210** | **0.158** | 0.385 | 0.381 |
| RGBD [18] | Pre-defined | Indoor | ✔ | ✔ | 0.352 | 0.453 | **0.129** | **0.286** | 0.345 | 0.339 |
| ScanNet [8] | Calibrated | Indoor | ✔ | ✔ | 0.480 | 0.496 | **0.126** | **0.246** | 0.367 | 0.365 |
| MVS [10] | Pre-defined | Indoor | ✔ | ✔ | 0.437 | 0.454 | **0.163** | **0.281** | 0.290 | 0.349 |
| Scenes11 [6] | Pre-defined | Synthetic | ✔ | ✔ | 0.451 | 0.445 | **0.168** | **0.410** | 0.381 | 0.383 |

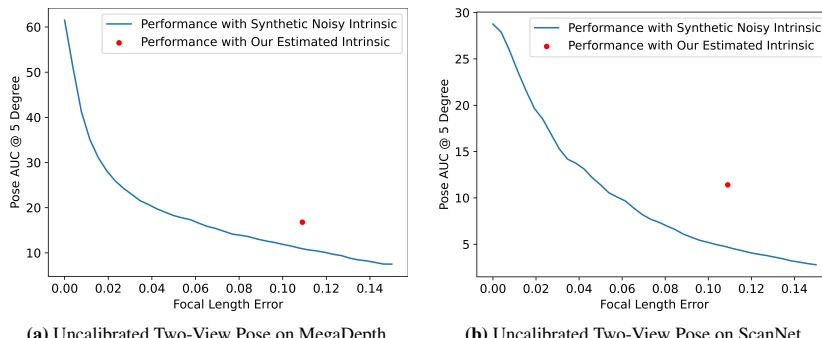

**(a)** Uncalibrated Two-View Pose on MegaDepth.  **(b)** Uncalibrated Two-View Pose on ScanNet.

**Figure 3:** We visualize the performance of uncalibrated two-view pose estimation with respect to the monocular camera calibration error $e_f$. The steep curve indicates that uncalibrated two-view pose estimation is a challenging problem. The red dot marks pose performance using our estimated intrinsic as reported in main paper Tab. 6.

to other training datasets. As a result, fitting the training distribution becomes easier, while fitting the testing distribution becomes more challenging. Considering the analysis provided above, it is considered reasonable to include the SUN3D dataset.

**In-the-Wild Monocular Calibration with Augmentation.** From Tab. 3, we synthesize novel intrinsic to **unseen** datasets following the augmentation defined in the main paper Sec. 4.1. Following main paper Sec. 3.6, the quality of the intrinsic can be assessed using the bounding box. In Fig. 4 to Fig. 8, we apply the same augmentation as Tab. 3 and visualize the intrinsic quality with bounding boxes. From Tab. 3, our focal length estimation shows superior robustness since the error $e_f$ remains consistent to the result without augmentation. Our focal point error $e_b$ goes up. However, in real-world applications, we consider the focal point error $e_b$ to be of lesser concern. Assuming a simple camera model naturally removes the focal point error $e_b$. In Tab. 3, we apply extensive augmentation, which breaks the simple camera assumption. But the assumption holds true in most real-world applications. Further, it is expected to have a high focal point error $e_b$. From Fig. 4 to Fig. 8, the focal length error $e_f$ indicates the correct restoration of the image aspect ratio. The focal point error $e_b$ indicates the accurate restoration of the cropping location. Intuitively, accurately locating the cropped area is a challenging task when applied to in-the-wild images. Meanwhile, as observed from Fig. 4 to Fig. 8, our model still relatively accurately locates the cropped area.

# 3 Downstream Applications

**Uncalibrated Two-View Camera Pose Estimation.** Fig. 3 plots the uncalibrated two-view camera pose estimation performance *w.r.t.* increasingly noisy intrinsic. We follow a naive way to synthesize noise into intrinsic:

$$f_x' = (1 + c \cdot e_f) \cdot f_x, \quad f_y' = (1 + c \cdot e_f) \cdot f_y, \tag{4}$$

where $c$ is a random sign whose value is either 1 or $-1$. Variable $f_x'$, $f_y'$, $f_x$, and $f_y$ are noisy and groundtruth focal length in axis X and Y respectively. From Fig. 3, the uncalibrated pose performance is highly sensitive to intrinsic noise, suggesting itself a challenging problem. Just as the geometric matching community [9, 19, 27] employs two-view pose estimation to assess the quality of correspondences, uncalibrated two-view pose estimation can also be utilized to evaluate the quality of intrinsic parameter estimation.

**Additional Image Cropping and Resizing Restoration Results.** We visualize the unseen ScanNet, Waymo, RGBD, MVS, and Scenes11 datasets in Fig. 4, Fig. 5, Fig. 6, Fig. 7, and Fig. 8 respectively.

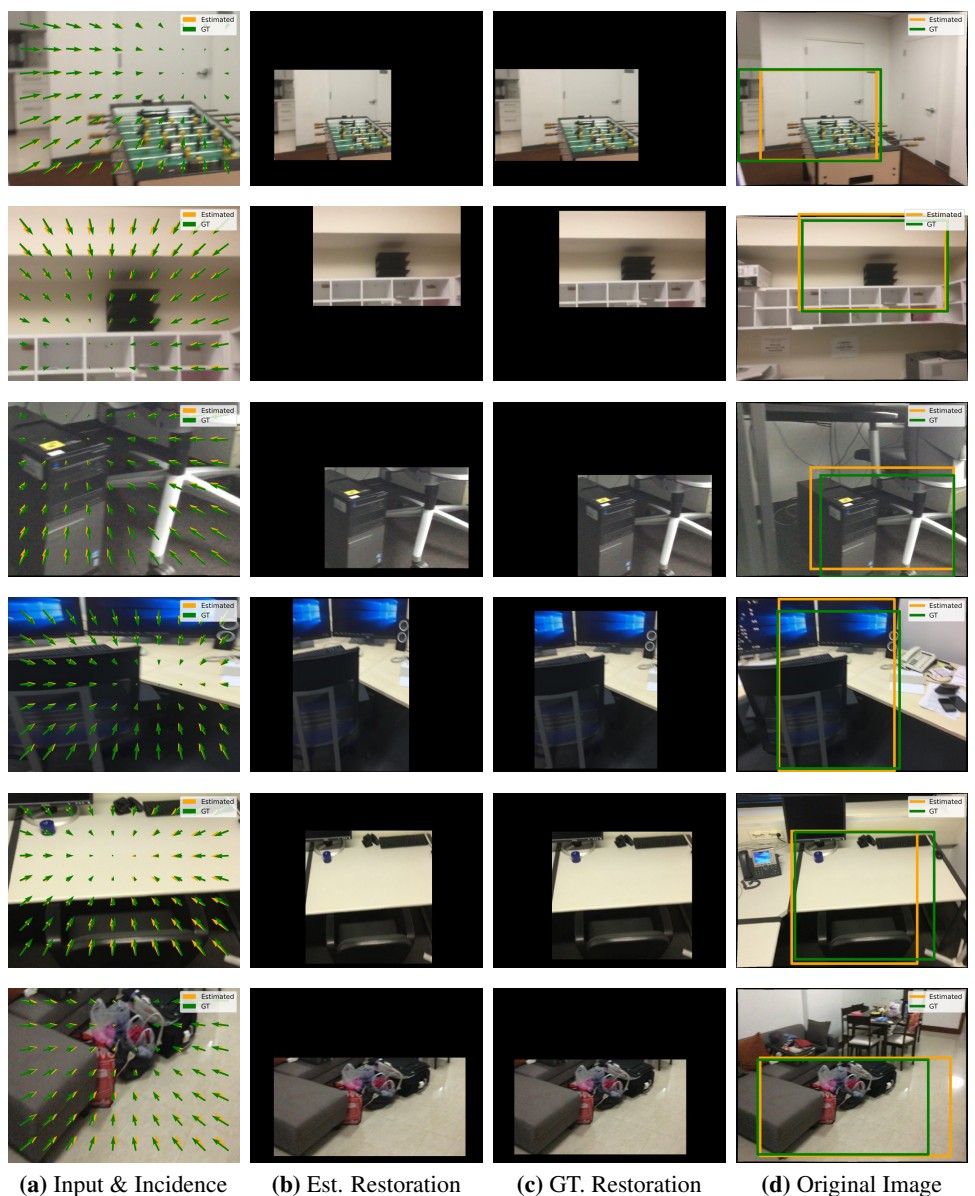

**(a)** Input & Incidence  **(b)** Est. Restoration  **(c)** GT. Restoration  **(d)** Original Image

**Figure 4: Image Crop & Resize Detection and Restoration Visualization on ScanNet.**

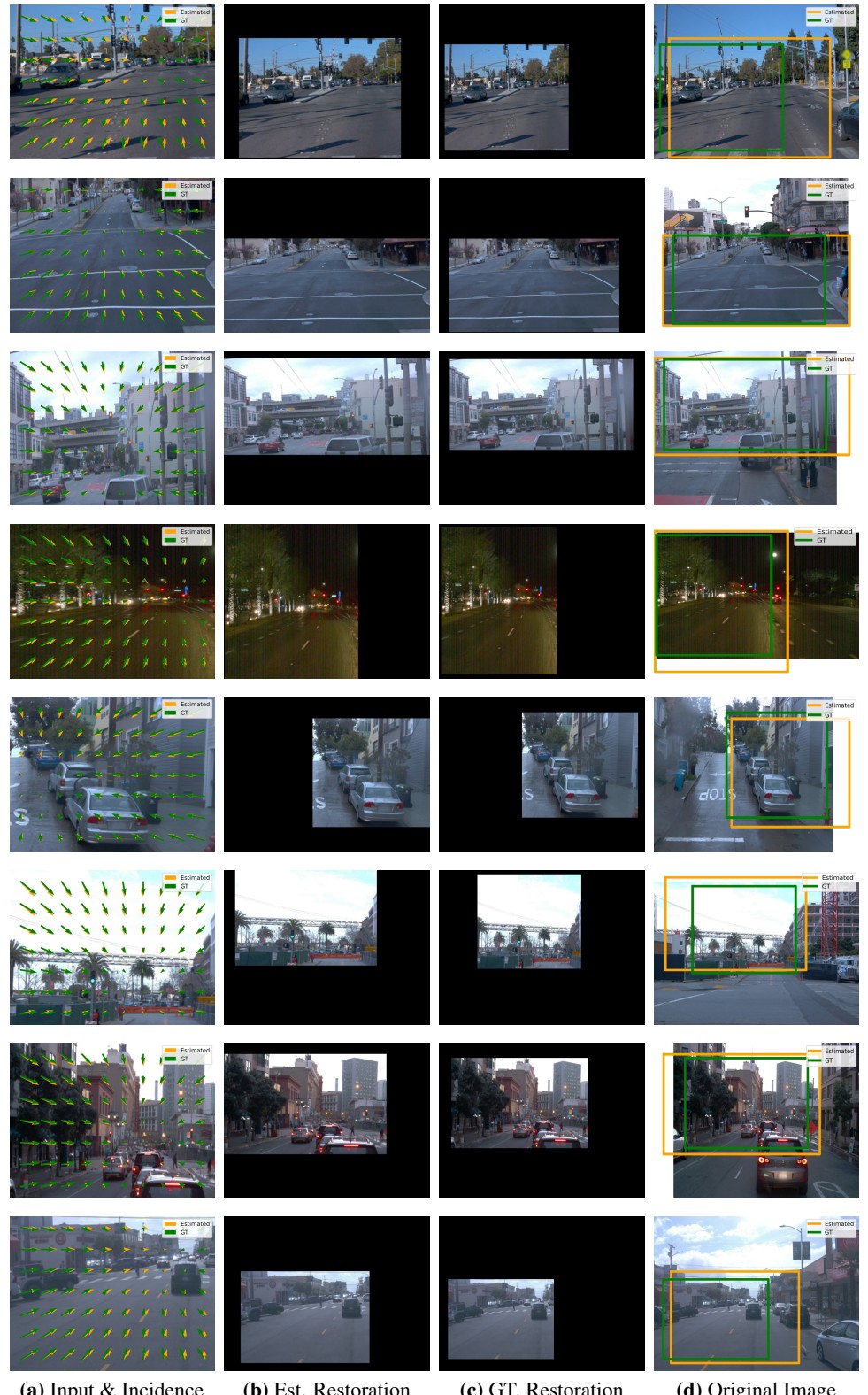

**(a)** Input & Incidence    **(b)** Est. Restoration    **(c)** GT. Restoration    **(d)** Original Image

**Figure 5: Image Crop & Resize Detection and Restoration Visualization on Waymo.**

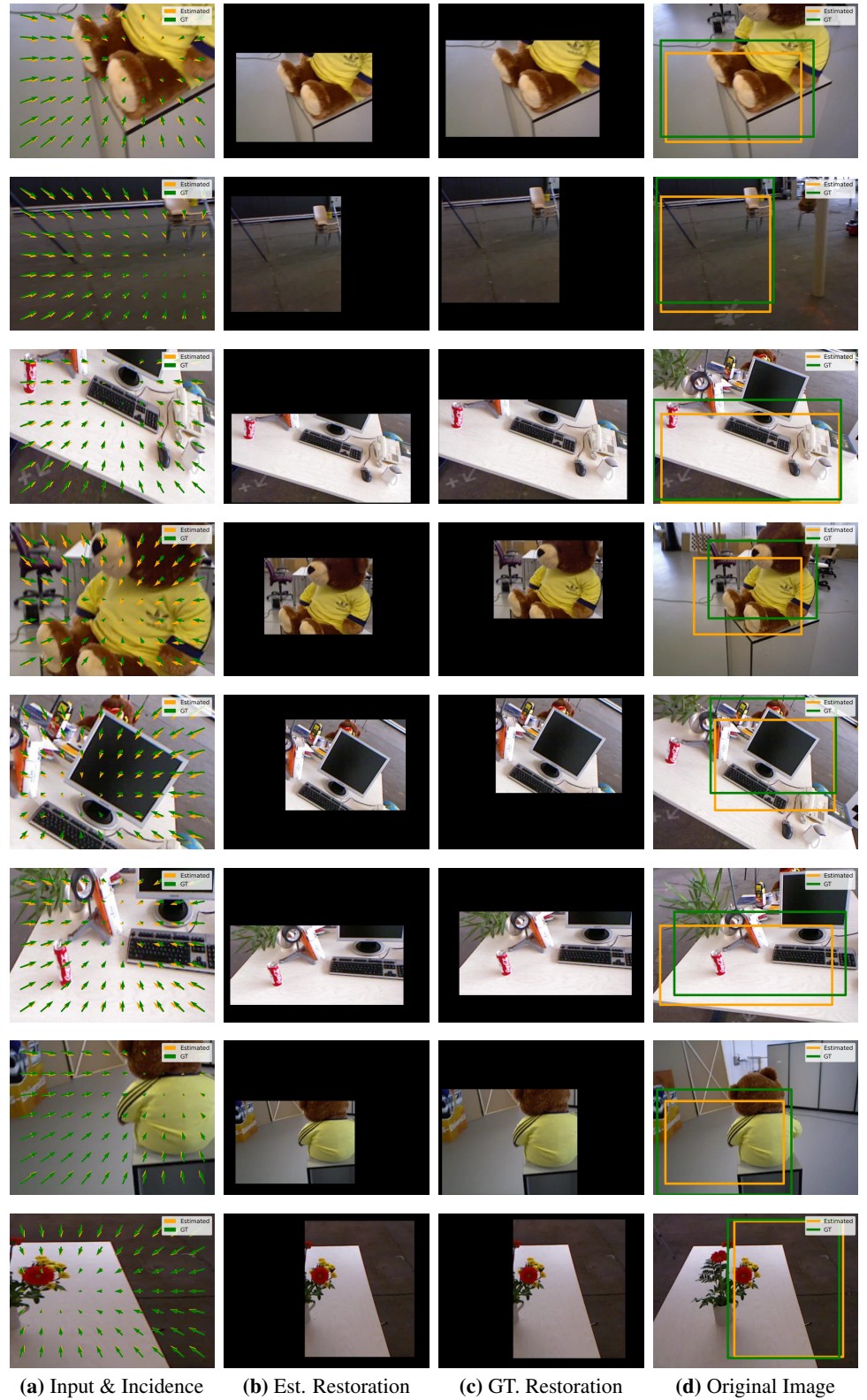

**(a)** Input & Incidence     **(b)** Est. Restoration     **(c)** GT. Restoration     **(d)** Original Image

**Figure 6: Image Crop & Resize Detection and Restoration Visualization on RGBD.** There exists overall image content due to the testing split only containing 160 images with overlapping content.

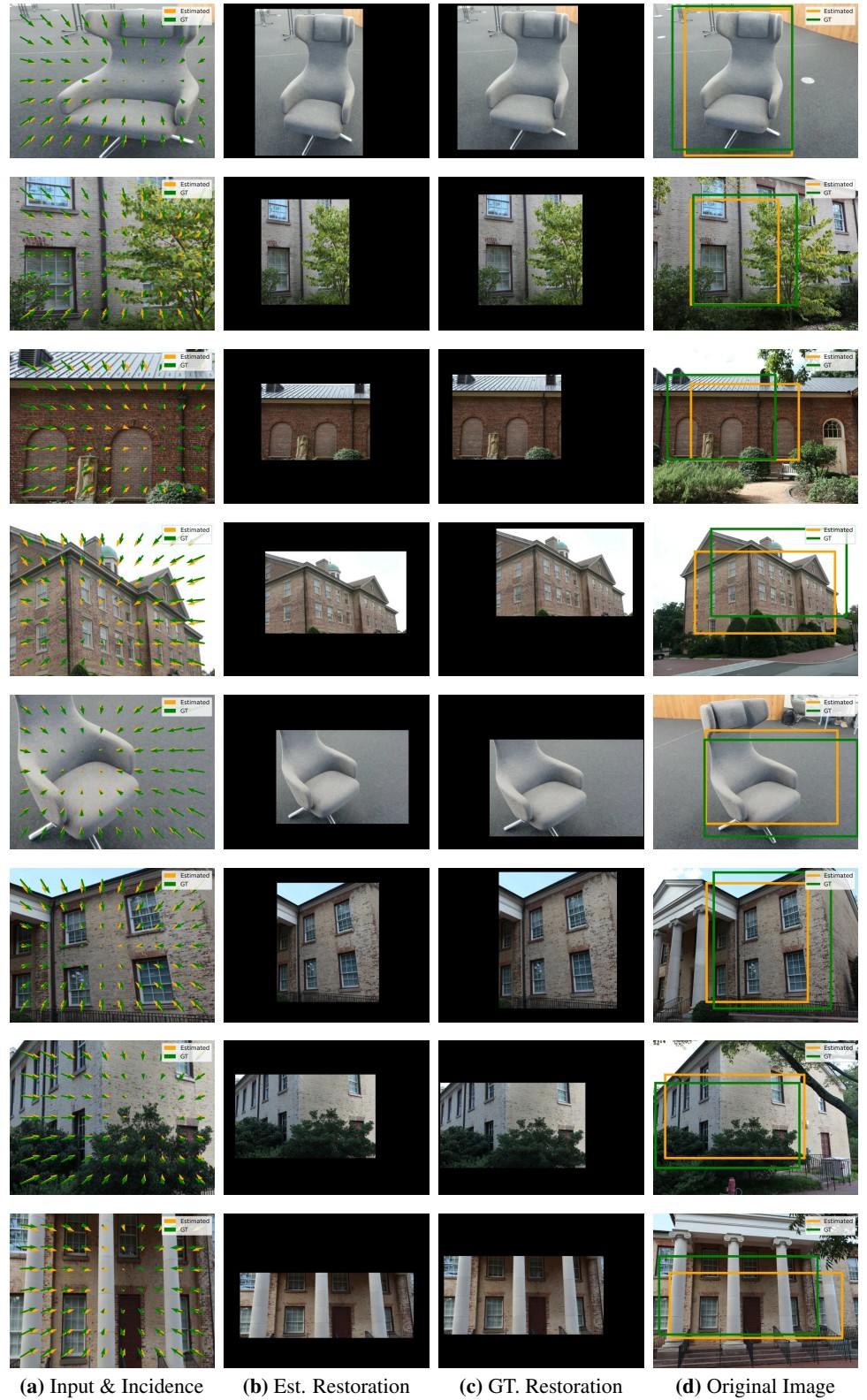

**(a)** Input & Incidence     **(b)** Est. Restoration     **(c)** GT. Restoration     **(d)** Original Image

**Figure 7: Image Crop & Resize Detection and Restoration Visualization on MVS.** There exists overall image content due to the testing split only containing 160 images with overlapping content.

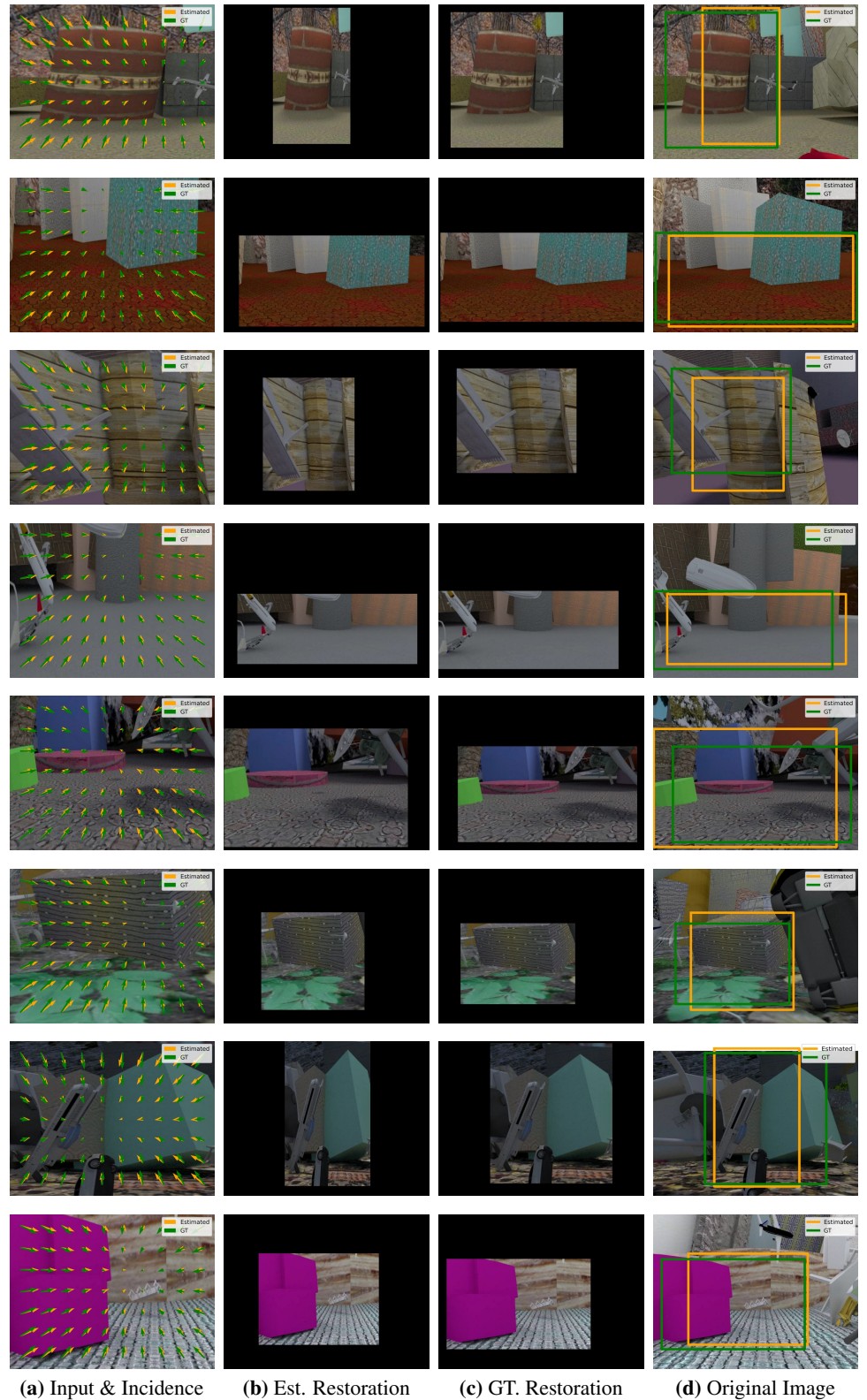

**(a)** Input & Incidence    **(b)** Est. Restoration    **(c)** GT. Restoration    **(d)** Original Image

**Figure 8: Image Crop & Resize Detection and Restoration Visualization on Scenes11.**