# OpenReview forum: "Tame a Wild Camera: In-the-Wild Monocular Camera Calibration"
_NeurIPS.cc/2023/Conference — NeurIPS 2023 poster_

### Official Review · Reviewer_iokN · 2023-06-24

**Soundness:** 3 good
**Presentation:** 2 fair
**Contribution:** 2 fair
**Rating:** 5
**Confidence:** 5

**Summary:**

The submission #3482, entitled "Tame a Wild Camera: In-the-Wild Monocular Camera Calibration" presents a novel self-calibration strategy where the regression of the incidence field of the camera is predicted via a deep neural network before using a RANSAC to filter outliers and regress the intrinsic parameters of the camera.

**Strengths:**

- Without being impeccable, the paper conveys the main idea behind the proposed technique.
- Combining the incidence field with more traditional methods to regress the camera intrinsic is "relatively" novel.
- The approach is supposed to work even when the images are cropped

**Weaknesses:**

- The literature is quite incomplete
- The camera distortion is not considered
- The approach is not really novel, as incident field regression has already been done in the past

**Questions:**

- 1. One of my main grips on this paper is the claim for novelty. While the reviewer should admit that using the monocular normal and the depth prediction to estimate the camera's intrinsic is new. The "incidence field" regression is not novel as it has already been described in "Deep Single Image Camera Calibration with Radial Distortion, CVPR, 2019". In this paper, the authors obtained mixed results using such a strategy. However, one of the significant advantages is to be able to deal with optical aberration, which can be challenging to model parametrically. Unfortunately, this manuscript does not integrate camera distortion, which would be meaningful in this context. I believe that introducing this non-parametric approach, is not fully justified, maybe by training [24] on the exact same cropped data would lead to similar results? (I have not seen its detail in the experiment part).
- 2. In this regard, it would also be beneficial to integrate "DeepCalib: A Deep Learning Approach for Automatic Intrinsic Calibration of Wide Field-of-View Cameras, CVMP 2018" in the literature as it deals with distortion and also explicitly tested on cropped images (maybe in supplementary). In their case, the estimation on cropped images fails because they have yet to train on these particular cases. So it may support your narrative. Another missing paper in the literature is the recent T-PAMI "A Perceptual Measure for Deep Single Image Camera and Lens Calibration" which might contain some interest in recent developments on deep learning-based calibration of cameras from a single image.
- 3. Would it be possible to also test on non-cropped images too? I feel it is unfair to test with many cropped images as, in reality, most images in the wild are uncropped
- 4. Since the approach has been trained on perspective images only, what would happen in case of a large field of view camera with a field of view outside of the training set distribution? In this regard it would be good to have some statistics regarding the distribution of the field on view used for the training.
- 5. To really work in the wild, integrating radial distortion seems necessary, how would you integrate it in your framework for training and inference?

**Limitations:**

I have already expressed all my concerns regarding this work in the previous sections of this review. For all the reasons above mentioned, I would like to issue a rather mixed opinion regarding the acceptance of this work.

---

> ### Author Rebuttal · Authors · 2023-08-10
>
>
> **Response to R5 ioKN**
>
> We sincerely thank R5 for the detailed comments. We deeply appreciate your feedback and aim to address your concerns thoroughly.
>
> Before responding to the concerns, we believe that there could be a possibility of misunderstanding in our approach:
> Different from [24, 26, 30, 31], [Deep Single], and [DeepCalib], our approach does not need perspective images in training. As in the main paper Tab. $1$, our approach avoids camera perspective calibration for intrinsic estimation. Except for Tab. $3$, all experiments use non-perspective images in training.
>
> We next delve into concerns in detail.
>
> **R5, Q1 Novelty of Incidence Field.**
>
> Thanks for bringing up the related work [Deep Single]! There exist several differences.  We summarize in Tab. D.
>
> Table D: Comparison of methods
>
> Method | Intrinsic DoF | Persp. Images Reqd | Network Outputs |
> -------|----| ------------ | ----|
> Deep Single | 1 | Yes | $4 \times 1$ Vector|
> Ours | 4 | No | $H \times W  \times 3$ Incidence Field|
>
> We elaborate on the differences further:
>
> 1. On Intrinsic Parameterization:
>
> 	[Deep Single] does not parameterize intrinsic as the incidence field. Instead, they parameterize the intrinsic as $1$ DoF camera vertical FoV (see [Deep Single] Fig. $2$).
>
> 2. On Network Inference:
>
> 	 [Deep Single] regresses $1 \times 4$ vector. We regress the $H \times W \times 3$ pixel-wise incidence field. Next, we use a RANSAC algorithm to retrieve the $4$ DoF intrinsic.
>
> 3. On Training Data:
>
> 	[Deep Single] trains with perspective images. Our method trains with perspective and non-perspective images.
>
> 4. Potential Confusion caused by [Deep Single] Bearing Loss:
>
>   The confusion possibly arises at the Bearing Loss in [Deep Single], depicted in Fig. 5. Our incidence field learning loss Eqn. 13 shares some similarities with the Bearing Loss, with the distinction that the latter requires camera perspective angles in computation. We acknowledge the similarity.
>   However, incidence field learning itself takes a brief paragraph in our work. Our novelty is more on parametrizing intrinsics as incidence field, and the accompanied RANSAC algorithm. The similarity does not impact our novelty.
>
> 5. Train [24] on cropped images.
>
> 	[24] parametrizes intrinsic as $1$ DoF vertical FoV. This limits [24] training on cropped images since cropped images have $4$ DoF camera intrinsic.
>
> **R5 Q2&5: Integration of Image Undistortion.**
>
> We appreciate this exciting suggestion of discussing image undistortion works.  We will update our reference to include the suggested three pieces of literature.
>
> We omit image undistortion within the scope of our paper based on three reasons:
>
> 1. Based on our observation, most public datasets and images on the internet are undistorted. Thus, most monocular camera calibration works, including all compared baselines ([25][30][24][31][26]), assume undistorted images as input.
> 2. Excellent solutions in learning-based image undistortion already exist. e.g., [Blind Geometric Distortion Correction on Images Through Deep Learning, CVPR, 2019].
> 3. In our setting, the image undistortion becomes an open-classification problem, where whether an image is distorted needs to be verified first. However, we consider this a non-trivial problem requiring another elaborate algorithm to solve and is currently beyond the scope of this paper.
>
> **Integrating Image Undistortion**
> We do agree that integrating the image undistortion enhances the generalizability of our model.
> Hence, we  discuss the following potential solutions for a distorted image:
>
> 1. Synthesize image distortion in the training dataset.
>
> 2. Estimate image distortion as a coordinate remapping similar to [Blind], together with the incidence field. This suggests estimating a pixel-wise vector:
> $[v_1, v_2, v_3, \Delta x, \Delta y]$, where $[v_1, v_2, v_3]$ is the incidence field, and $[ \Delta x, \Delta y]$ is the distortion flow following [Blind].
>
> 3. Recover image distortion using Hough Transform following [Blind].
>
> 4. Recover the camera intrinsic using the updated coordinates with distortion corrected.
>
> **R5, Q3 Test on uncropped images.**
>
> We test primarily on uncropped images:
>
> - **All** test images in Tab. $3$ are uncropped.
>
> - **Lower** half of Tab. $2$ tests real uncropped images from datasets such as MegaDepth, Waymo, RGBD, ScanNet, MVS, and SceneNet.
>
> **R5, Q4 Statistics and performance of camera FoV.**
>
> **FoV Statistics** We report our training/testing data FoV statistics Fig.$3$ of the Supp material.
> The figure shows that our training data primarily includes $30-80$ degrees horizontally and $15-60$ degrees vertically.
> Further, our training and testing data does contain some particularly large (over $60$ degrees) or small FoV (below $20$ degrees) images.  Most of our training data is not perspective images.
>
> **Performance Statistics** We thank you for suggesting this insightful analysis!
> Therefore, we recompute Tab.$2$ results by varying the horizontal FoV ranges to obtain Tab. E.
> We report the intrinsics performance without applying assumptions and by averaging over all datasets in each range.
>
> Table E: Intrinsics estimation results with FoV variation.
>
> |     FoV     | 10 - 20 | 20 - 30 | 30 - 40 | 40 - 50 | 50 - 60 | 60 - 70 | 70 - 80 | 80 - 90 |   All-Range  |
> |:-----------:|:-------:|:-------:|:-------:|:-------:|:-------:|:-------:|:-------:|:-------:|:------:|
> |    Number   |   163   |   726   |   1927  |   2747  |   2554  |   404   |   151   |    21   |  8708  |
> | Percentile  |   1.9%  |   8.3%  |  22.1%  |  31.5%  |  29.3%  |   4.6%  |   1.7%  |   0.2%  | 100.0% |
> |    $e_f$    |  0.173  |  0.119  |  0.113  |  0.120  |  0.128  |  0.109  |  0.120  |  0.112  |  0.122 |
>
> Tab. E suggests that our algorithm maintains robustness across various FoVs, with the exception of exceptionally small ones. This outcome is anticipated, as small FoVs inherently offer a highly limited camera view.

---

> > ### Comment · Reviewer_iokN · 2023-08-14
> >
> > Thank you for your clear rebuttal. I apologize for any misconceptions in my original review. Your proposed approach seems indeed to differ significantly from previous works. Although I agree with R2 that this paper would be better suited for a computer vision conference, I would like to maintain my initial positive rating.

---

> > > ### Author Response · Authors · 2023-08-14
> > >
> > > We are grateful for your recognition of the novelty in our work. Your positive comments mean a lot to us. Thank you once again for dedicating your time and effort to reviewing our paper!

---

### Official Review · Reviewer_Ujoz · 2023-07-01

**Soundness:** 4 excellent
**Presentation:** 3 good
**Contribution:** 3 good
**Rating:** 6
**Confidence:** 4

**Summary:**

This paper proposes a method for single-image camera calibration using a 3D prior the authors refer to as an incidence field. The incidence field is the collection of rays originating from some 3D point towards the camera origin, that are incidental to the image plane. The authors describe how the incidence field can be used to recover the camera intrinsics. Next, a method is proposed for recovering the intrinsics by first estimating an incidence field from an image (using a recently introduced neural network architecture, NewCRFs) and then applying RANSAC. An extensive evaluation shows that the proposed method achieves superior performance to recent methods on numerous benchmarks.  Additionally, several example applications are demonstrated.

**Strengths:**

- Addresses an important and interesting problem. Single image camera calibration "in the wild" has received a lot of attention lately and is a fundamental vision task. Knowledge of camera intrinsics is essential for numerous applications.

- The proposed approach has technical novelty. In particular, shows how the concept of an incidence field (backprojected rays in 3D space) are related to camera intrinsics, then introduces a method to estimate them (method of [62] plus RANSAC). As far as I know, this has not been done before. Often in this space, a novel take on how to represent something leads to downstream performance benefits.

- The method is extremely simple, which I consider a positive.

- The evaluation is extensive, achieving superior results to recent methods on several benchmarks.

- Numerous interesting applications presented, such as detecting image resizing and cropping.

- Code provided and will be released with data and models.

**Weaknesses:**

- The evaluation should have descriptions of the metrics and how they are calculated (found in supplemental).

- The quality of the writing is poor in certain aspects. For example, L14 in the abstract: "With the estimated incidence field, a robust RANSAC algorithm recovers intrinsic." Ultimately, the manuscript just needs a solid editing pass top-to-bottom.

- The quality of the figures, though parseable, could be improved from an aesthetic sense.

**Questions:**

My initial rating is a weak accept. The proposed approach is simple, explained well, has novelty, and the results are compelling. I think this will be of interest to the community.

Suggestions:

- Full editing pass to address minor language issues. Improve aesthetics of figures (e.g., Figure 2, left).

- Implementation details should be in the main document instead of supplemental.

- L233 should reference Table 3. Text should introduce the dataset.

**Limitations:**

Limitations minorly touched on in the manuscript.

---

> ### Author Rebuttal · Authors · 2023-08-10
>
>
> **Response to R4 Ujoz**
>
> We appreciate the reviewer's suggestions to make our paper accessible to a wider audience.  We extend special gratitude to the reviewer for recognizing the technical novelty of our approach and its potential contribution to the community. We will adhere to the reviewer's suggestions in the revision.
>
> **R4, Q1 Missing metrics in tables.**
>
> Thanks for the suggestion! We will move the evaluation metrics and implementation details from the supplementary to the main paper.
>
> **R4, Q2 Improve the quality of writing.**
>
> Although we have partially revised our manuscript based on the reviewer's feedback, we will do a solid editing pass over the paper.
>  We include a list of revisions made so far. We will continue improving the manuscript.
>
> - L$14$: ''With the estimated incidence field, a robust RANSAC algorithm recovers intrinsic.'' -> "We apply a robust RANSAC algorithm to recover intrinsic from the estimated incidence field."
> - Fig.$2$c, "w/ Assum." to "w/o Assum.", and change the other one accordingly.
> - Tab.$3$ "erceptual" to "perceptual"
> - L$233$, Tab.$4$ reference to Tab.$3$ reference.
> - Additional description of Tab. $5$ and Tab. $6$ baselines.
>
> **R4, Q3 Quality of Fig. 2.**
>
> We re-plotted Fig. $2$, improving its layout and removing the ``L" shape illustration boxes. We do not place this in the one-page rebuttal pdf because of space constraints.
>
> **R4, Q4 L233 should reference Table 3. Text should introduce the dataset.**
>
> We fixed these issues in our manuscript.

---

> > ### Comment · Reviewer_Ujoz · 2023-08-15
> >
> > Thank you for your response. I have read the other reviews and author rebuttals. The majority of reviewers are in agreement and I will be retaining my initial rating.

---

### Official Review · Reviewer_PGSe · 2023-07-04

**Soundness:** 3 good
**Presentation:** 4 excellent
**Contribution:** 4 excellent
**Rating:** 7
**Confidence:** 5

**Summary:**

This paper introduces incidence field, a per-pixel representation for single image camera intrinsics calibration. The incidence field is defined as a 2D vector, pointing to the principal point of the image, normalized by the focal length. It is invariant to cropping and resizing. The paper suggests using a neural network to regress the incidence field given an input image. Finally, RANSAC is used to get the 4DoF intrinsics (fx, fy, cx, cy).

**Strengths:**

The paper connects the incidence fields with surface normals and depthmaps. Thus, making it a reasonable representation that can be predicted by a network such as NewCRF, which is originally designed for depth estimation.

The incidence field formulation is simple yet powerful. It has desirable properties such as invariant to arbitrary cropping and resizing, as shown in S3.3; and can be derived from surface normals and depth, as shown in S3.2.

Incidence fields are independent of camera extrinsic, contrasting Perspective Fields [26], which requires camera roll and pitch for formulation. This feature enables incidence fields to be trained on extensive data where the camera is calibrated but the pose is unknown.

The paper demonstrates SOTA performance on various datasets, both qualitatively and quantitatively.


**Weaknesses:**

- The paper claims in its contribution that “Our method meaks no assumption for the to-be-calibrated image”, while in fact, it presumes no distortion in the image (L118).

- Table 1 needs revision, as it inaccurately describes the baselines and the comparisons are not apple-to-apple. Contrary to the claims made, [24, 26] do not assume Manhattan data during training and can be trained on arbitrary scenes, such as natural scenes. Furthermore, it is unfair to state that your method requires no assumption on the training data when comparing it with others [24, 31, 26], since they also predict camera roll and pitch while your method does not.

- There are missing details in Table 2. It is unclear what the unit for the error metric is, and which dataset [26] is trained on. If [26] is solely trained on GSV data, then it implies a zero-shot scenario for [26] on all the listed datasets in Table 2. If camera extrinsics are provided, you should be able to train [26] on the same training set as your method, ensuring a fair comparison.

- Including a baseline that predicts intrinsics from depth and surface normal predictions (as described in Sec 3.2) would be beneficial. This would empirically validate the derivation and give an idea of the method’s accuracy in Sec 3.2. Furthermore, it would support the argument in L150 that “Minimal solver in Eq. (9) can lead to a poor solution”, therefore further justifying the contribution of the method in Sec 3.4.

- Typo in L218 and L233: There is no [26] in Tab 4. It should be Tab 3 instead.

- In Table 5 and Sec 4.4, the baseline method is not clarified.


**Questions:**

See weakness.

**Limitations:**

In real application, whether to apply the assumption still waits for human input.

---

> ### Author Rebuttal · Authors · 2023-08-10
>
>
> **Response to R3 PGSe**
>
> We sincerely thank for the exceptional comments and for giving our paper an "Accept." We are particularly grateful for recognizing the validity of parametrizing intrinsic as an incidence field. We next delve into your specific feedback.
>
> **R3, Q1 Undistorted Image Assumption.**
>
> Great Catch! We apologize for the imprecise argument. We assume an undistorted image and will clarify our claim and update the abstract, introduction, related works, and Table $1$ in the revision.
> However, we humbly suggest that the undistorted image assumption is commonly upheld in all compared baselines ([25][30][24][31][26]), particularly in line with the Manhattan World assumption [31, 32, 47, 63], where detecting image lines inherently relies on an undistorted image.
>
> **R3 Q2&3: Fair Comparison in Tabs. $1$ and $2$.**
>
> There is a misunderstanding in baselines. Baseline [26] and other baselines [24, 30, 31] can **NOT** be trained using ground truth camera extrinsics. Instead, they assume a gravity-aligned camera during training.
> The gravity direction is unknown in most public datasets, such as ScanNet.
> In other words, the known extrinsic only establishes the camera pose relative to the first camera.
> In contrast, the orientation of the first camera relative to gravity remains unknown.
> Now, we address the concerns comprehensively:
>
> 1. **Manhattan data assumption in training.**
>
> Thank you for bringing this to our attention!
> The Manhattan World assumption eventually gives the camera's roll, pitch, and focal length. We illustrate the process briefly below:
>
> 	Manhattan World assumption
> 	↳ Parallaized Line Sets,
> 	↳ Two (or three) Vanishing Points,
> 	↳ Camera's roll, pitch, and focal length.
>
> Recent learning-based methods [24][31][26] directly acquire the groundtruth camera's roll, pitch, and focal length from panorama images. We term this as the 'Manhattan data assumption,' which ultimately obtains groundtruth derived from the Manhattan World assumption. It does not imply the presence of horizontal and vertical lines in the images. We will further clarify this point with an additional description in our Tab. $1$ and Sec. $2$.
>
> 2. **Unfair comparison to [24, 31, 26] in Tab. $3$.**
>
> Great point! We view Tab. 3 as a fair comparison since our reformulation eliminates the need for perspective calibration. Following the Manhattan World illustration above, traditional baseline methods [31, 32, 47, 63] require simultaneous perspective calibration and intrinsic calibration. This also applies to the SoTA learning-based methods. For instance, [26] necessitates line detection before camera intrinsic estimation, while [31] estimates camera intrinsic using an inferred camera perspective field. In short, perspective calibration is indispensable for camera calibration in baselines.
>
> Next, we extend our method and conduct experiments to jointly apply perspective and intrinsic calibration to address your concern in Tab. $B$ of the one-page rebuttal pdf.  We modify our network to jointly regress the incidence and perspective fields (defined in [26]).
>
> Interestingly, with the incidence field estimated, we can directly estimate the camera roll and pitch from the perspective field. The incidence vector $\mathbf{v}$ and perspective up vector $\mathbf{u}\_x$ (Eqn. $1$ of [26]) determine a 3D plane where the gravity vector (cross-focal point) resides. The intersection of two such planes in 3D space gives the gravity direction. We solve gravity using this as a minimal solver with a RANSAC algorithm. Finally, we determine the horizon line by averaging over $\varphi_{\mathbf{x}}$ (Eqn. $2$). Tab. B shows that our method also outperforms the SoTA methods in perspective calibration.
>
> 3. **Unfair comparison to [26] in Tab. $2$.**
>
> We now compare with the most recent checkpoint (released after the NeurIPS submission deadline) in Tab. $A$ in the one-page PDF.
> The newer released checkpoint [26] was trained on bigger 360Cities and EDINA datasets. However, we cannot train our model on 360Cities since the dataset is not public. From Tab. $A$, our method maintains SoTA performance with a solid margin over [26].
>
> **R3, Q4 Sec. 3.2 baseline.**
>
> Thank you for your valuable input! In the one-page rebuttal PDF Tab. $A$, we incorporate a baseline with Sec. 3.2 minimal solver.
> We utilize ZoeDepth and [Estimating and Exploiting the Aleatoric Uncertainty in Surface Normal Estimation, ICCV, 2021] to initialize monocular depthmap and surface normal. These experiments further justify our idea of parametrizing intrinsic as the incidence field.
>
> **R3, Q5 Typos.**
>
> We appreciate for pointing out the typos! We have corrected them in the revised version.
>
> **R3, Q6 Baselines in Tab. $5$ and Sec. $4.4$.**
>
> Apologies for the oversight.
> The baselines in Tab. $5$ adopt an identical network structure, with only the last layer modified to a fully connected layer after average pooling for direct regression of the $4$ degrees of freedom (DoF) intrinsic. Additionally, all these models normalize the intrinsics by resizing the image height and width to [0, 1], which aids model convergence. We compare the baseline and our method’s estimated intrinsics using the same evaluation protocol.
>
> The baselines in Tab. $6$ of Sec. $4.4$ follow LoFTR in estimating image correspondence. With the estimated correspondences, they apply an OpenCV-based five-point algorithm with RANSAC to estimate the two-view camera pose.

---

> > ### Comment · Reviewer_PGSe · 2023-08-12
> >
> > Hello, thanks for the great effort in the rebuttal.
> >
> > The new results in the pdf resolve my concern. Thanks for the updated experiments to ensure a more fair comparison.
> >
> > Here are some nitpicking comments in response to the rebuttal:
> > -  In my opinion, instead of using "Manhattan world assumption" to describe the requirements for some baselines, "gravity-aligned panoramas" is a more accurate term. I still can't entirely agree that previous baselines require a "Manhattan World assumption". Manhattan World assumption is that "all surfaces in the world are aligned with three dominant directions, typically corresponding to the X, Y, and Z axes;" (Furukawa, Yasutaka, et al. "Manhattan-world stereo." CVPR 2009.) Getting roll, pitch, and focal length from panorama images [24, 26] only requires the panorama to align with gravity, which is aligning only one axis.
> >
> > - "For instance, [26] necessitates line detection before camera intrinsic estimation, while [31] estimates camera intrinsic using an inferred camera perspective field." In this sentence, [26], [31] should be swapped.

---

> > > ### Author Response · Authors · 2023-08-13
> > >
> > > We greatly appreciate your rectification of the reference order for [26] and [31] in the rebuttal response. Regarding our discussion on the "Manhattan world assumption", your suggestion of substituting it with "Gravity-Aligned Panorama Images" is excellent! We plan to apply the following changes:
> > >
> > > - We will substitute the main paper Tab.$1$ from "Manhattan-Train" to "GravityAlignedPanorama-Train".
> > > - We will clarify the relationship between the "GravityAlignedPanorama-Train" and "Manhattan World assumption". Both assumptions yield the camera roll, pitch, and focal length. However, the former accommodates natural scenes where surfaces are not necessarily aligned with principal axes.
> > >
> > > We believe there's a specific point that may need further discussion:
> > >
> > > - "Manhattan World Assumption" suggests the imaging content (surfaces) is aligned to X, Y, and Z axes.
> > > - "Gravity-Aligned Panorama Images" suggests the imaging plane is aligned to the gravity (X axes).
> > >
> > > We think the two are not directly comparable as one is about imaging content and the other is about imaging plane. But we do agree "Gravity-Aligned Panorama Images" relaxes the "Manhattan World Assumption" as accommodating non-aligned surfaces. "Gravity-Aligned Panorama Images" is a more accurate description for the baselines [24, 26, 31].

---

### Official Review · Reviewer_wjnN · 2023-07-06

**Soundness:** 3 good
**Presentation:** 3 good
**Contribution:** 2 fair
**Rating:** 3
**Confidence:** 3

**Summary:**

An in-the-wild monocular camera calibration method is proposed. It allows to estimate the focal lengths $f_x$ and $f_y$ as well as the optical center $b_x$, $b_y$ without any additional information such as a checkerboard or the Manhattan world assumption. The proposed method consists in employing a neural network to predict the incidence field, followed by a RANSAC. Several experiments are performed to demonstrate the ability of the proposed method to perform in-the-wild calibration.

**Strengths:**

1. The proposed approach is simple and efficient.
2. Training a network to predict the incidence field seems straightforward (the authors took a depth prediction network architecture and simply changed the last layer and the loss).

**Weaknesses:**

1. The idea of using a sota depth prediction network to predict the incidence field (section 3.4) seems new, but this is a rather weak contribution. If I am not mistaken, it essentially consists in taking the network from a github page, change the last layer and the cost function and retrain the network on the same data. This contribution seems weak for a conference like NeurIPS.
2. The idea of using the incidence field to estimate the calibration parameters (section 3.5) seems new, but the derivation is trivial: here the minimal solver is the solution of a trivial linear system (it takes a few minutes to derive, compared to Grubner-basis-based minimal solvers that we encounter in essential matrix estimation). In terms of contribution, I believe this is not strong enough for NeurIPS. Btw, I believe eq.17 and 18 do not correspond to a RANSAC since you simply quantize the focal length space and test each value.
3. I believe section 3.2 is not a contribution, is it ? It is said that this calibration method will not produce good results but I could not find any experiment using it. I suggest to include this experiment.

**Questions:**

After carefully reading the paper, I recommend to reject it. I believe the contributions are not sufficient for a conference like NeurIPS and not interesting for the broader NeurIPS community. I suggest to submit the paper to another conference like 3DV or CVPR.

**Limitations:**

.

---

> ### Author Rebuttal · Authors · 2023-08-10
>
>
> **Response to R2 wjnN**
>
> We thank the reviewer for the positive comments on the soundness and presentation of the paper. We greatly appreciate the valuable, constructive criticism. Your concern helps us to further address the missing connection between Sec. $3.2$ and other sections in the manuscript. We address the concerns in detail.
>
> **R2, Q1: Using the SoTA depth network to predict the incidence field is a weak contribution.**
>
> We do not claim any contribution in learning the incidence field. Therefore, we briefly discuss learning the incidence field in a short paragraph (Sec.$3.4$) of our methodology section.
>
> **R2, Q2 - Part1: Monocular Camera Calibration with incidence field provides insufficient contributions to the community.**
>
> We restate our contributions in monocular camera calibration:
>
> 1. **Assumption-free method for undistorted in-the-wild images.**
> 	Monocular image-based 3D sensing is gaining rapid attention, and most of these sensing methods require known camera intrinsics. Therefore, an intrinsic estimation method for monocular images is the need of the hour. In Fig. A of the one-page rebuttal PDF, we enhance the monocular 3D object detection results of a recent CVPR 2023 work, Omni3D, by replacing their predefined intrinsics with estimated intrinsics from our method.
>
> 2. **First to infer the $4$ DoF intrinsics without extrinsics from monocular 3D prior.** Prior works always infer intrinsics alongside the camera extrinsics, either as camera pose ([25]) or perspective angles ([24, 26, 30, 31]).
>
> 3. **Simple and Neat solution with competitive performance**. Our method eliminates the need for extrinsic estimation in camera calibration, simplifying the algorithm and enhancing performance.
>
> **R2, Q3: Sec. $3.2$ is not a contribution.**
>
> We respectfully disagree. While the algorithm in Sec. 3.2 is not finally adopted and is not a contribution, the following insights we learned from this algorithm are contributions.
>
> - First, Sec. $3.2$ suggests the one can infer the intrinsics from monocular 3D prior.
>
> 	 To the best of our knowledge, we are the first to show this.
> 	 Prior works always infer intrinsics alongside camera extrinsics. Some use 3D-2D correspondence sets with a predefined 3D template [25] (Tab. $4$ baseline). Others jointly solve intrinsics with the camera roll and pitch angle [24, 26, 30, 31] (Tab. $3$ baselines). Sec. $3.2$ provides one solution which **avoids** extrinsic estimation by directly solving $4$ DoF intrinsic from monocular 3D prior. The disentanglement also simplifies the design of the minimal solver, representing a significant contribution of our work.
>
> - Second, Sec. $3.2$ suggests the incidence field is a 3D prior.
>
>   Proving the incidence field as a 3D prior is crucial, as the incidence field is not a typical 3D prior.
>   Monocular depthmap and surface normal are image texture dependent, which changes w.r.t. image content. In comparison, the incidence field is less texture dependent. For a given image with size $H \times W$, its incidence field is identical over different images. However, the incidence field still depends on image content. Given an image, after cropping and resizing, the incidence field changes accordingly, as the Fig. $3$ of the main paper. In Sec. $3.2$, Eqn. $4$ suggests the incidence vector is uniquely determined by the surface normal and monocular depthmap. The incidence vector $\mathbf{v}$ in Eqn. $4$ has $2$ DoF, uniquely determined by the $2$ constraints provided in Eqn. $4$. In summary, Sec. 3.2 suggests the incidence field is a monocular 3D prior, uniquely determined by the well-defined monocular 3D prior surface normal and depthmap. The fact implies that incidence field learning can generalize to different scenes in a similar way to surface normal and monocular depthmap.
>
> - Third, we make up the Sec. $3.2$ baseline performance in the Tab. A of the one-page rebuttal pdf. From Tab. A, the Sec. 3.2 minimal solver gives poor calibration results. This rationalizes the proposal of the incidence field for camera calibration. Thanks for suggesting this!
>
> **R2, Q2 - Part2: The proposed minimal solver is naive.**
>
> - First,  following **R2, Q3**, using the RANSAC algorithm to recover the intrinsics is just a part of our contribution.
>
> - Second, Our incidence field-based solution is neat, yet a benefit.
> 	From **R2, Q3**, our incidence field-based solution is neat as it avoids extrinsic estimation. Further, our neat solution facilitates rapid implementation and strong performance. Together with our efforts in creating a dataset and defining evaluation metrics, our work serves as a benchmark for other in-the-wild monocular image calibration works. Please also see the comments from the **R4**, who acknowledges the benefit of a neat solution.
>
> Regarding the Eqn.$17$ and $18$,  we will modify the subtitle in L$179$ to "Enumerate w/ Assumption". Thank you for pointing this out!

---

> > ### Comment · Reviewer_wjnN · 2023-08-18
> > **Response**
> >
> > Hello,
> >
> > Thank you for your answers.
> >
> > As far as I am concerned, as I explained in my initial review, even if the paper is technically sound and the results are good, I strongly believe the contributions are not sufficient for a conference like NeurIPS.
> >
> > However, I can see that several reviewers are excited by this paper. Maybe I am missing something... As a consequence, I will keep my initial rating but I will not fight against the other reviewers if they decide to recommend to accept the paper.
> >
> > Best regards,
> > wjnN

---

### Official Review · Reviewer_TGwc · 2023-07-08

**Soundness:** 4 excellent
**Presentation:** 3 good
**Contribution:** 4 excellent
**Rating:** 7
**Confidence:** 3

**Summary:**

The authors propose a novel learning based approach for 4 DOF camera intrinsics calibration from a single image in the wild. They propose to use a neural network to predict the incidence rays for each pixel in an image from which the camera intrinsics can be recovered. The authors motivate the prediction of the incidence rays as an alternative for predicting the depth and normals, as the computation of the intrinsics from these signals involves derivatives it is very succesible to noise. The evaluation of the authors shows great potential of this method to generalize to images in the wild and show potential applications like uncalibrated 2-view pose estimation and image transformation detection.

**Strengths:**

The idea of leveraging the relationship betwen the scene depth and normals and the incidence rays is very interesting.
The explanations are great and the neccesary derivations are all there. There are comparisons to both monocular methods using geometry and methods using known objects.

**Weaknesses:**

Most of the tables (2, 4 & 6) with the quantitative results do not provide a metric unit and while the results are compared to existing methods, the metrics should be introduced properly. For 4.4 and table 5 the description and or reference of the baseline is missing.

**Questions:**

Please add the metrics to the tables, introduce them and provide a reference for the definition.
Add the description of the baseline in 4.4.
How does the performance depend on the image content and the sampling of pixels? E.g. what samples are usually providing the best hypotheses and what would intuitively be a minimal image content that would allows the model to predict the incident rays? The same as for depth and normal prediction?

**Limitations:**

Similar to depth and normal prediction there are probably cases where the image content is not sufficient to predict incidence rays. The authors did not discuss this case as far as I can see.

---

> ### Author Rebuttal · Authors · 2023-08-10
>
>
> **Response to R1 TGwc**
>
> We sincerely appreciate the valuable feedback provided by the reviewer to enhance our manuscript quality. We also agree on the importance of properly introducing metric units in tables and providing clear references for baselines. We will address these aspects appropriately. We offer the following responses to the raised questions:
>
> **R1, Q1: Missing table metric units.**
>
> Thank you for bringing this to us!
> Tabs. $2$ and $4$ evaluation metrics are at L$31$ of the supplementary.
> For Tab. $6$, we follow the evaluation protocols of baseline LoFTR. The reported AUC of the pose error is measured at thresholds (5°, 10°, 20°), with the pose error defined as the maximum angular error in rotation and translation. Please refer to Sec. $4.2$ of LoFTR for the detailed evaluation protocols.
> We have revised our main paper to incorporate these metrics.
>
> **R1, Q2: Missing reference to Sec. $4.4$ and Tab. $5$ baselines.**
>
> We apologize for the missing details. In Tab. $5$, the baseline shares an identical network structure, with the last layer modified to a fully connected layer for direct regression of the $4$ degrees of freedom (DoF) intrinsic. During training and testing, we normalize the intrinsic matrix by resizing the image height and width in the range [0, 1], which aids model convergence.
> We compare the baseline and our method's estimated intrinsics using the same evaluation protocol.
>
> The baselines in Tab. $6$ of Sec. $4.4$ follow LoFTR in estimating image correspondence. With the estimated correspondences, they apply an OpenCV-based five-point algorithm with RANSAC to estimate the two-view camera pose.
>
> **R1, Q3: Calibration performance w.r.t. sampling strategy.**
>
> That is a very insightful question! Monocular camera calibration eventually relies on image projective distortion. This leads us to think sampling around image edges could enhance results, as projective distortion is most discernible near borders. We run the following ablation to verify the reviewer's insight.
>
> With normalized image coordinates ranging from [-1, 1], we ablate calibration performance through sampling: initially around the image border and then spanning the entire image. We define the sampling area w.r.t. threshold $k$ as:
>
>  {$ (x, y) \mid  |x| \geq k, |y| \geq k, x \in [-1, 1], y \in [-1, 1] $}
>
> where $k$ closer to $1$ refers to sampling around the image edges, while $k=0.0$ spans the entire image.
>
> We evaluate the MegaDepth, ScanNet, and Waymo datasets, reporting calibration performance with the simple camera assumption applied in Tab. C. The performance at $k=0.0$, thus, aligns with the results in Tab. $2$ of the main paper.
>
> Table C. Calibration performance with sampling strategy.
>
> | $e_f$ \ $k$ |  0.7  |  0.6  |  0.5  |  0.4  |  0.3  |  0.2  |  0.1  |  0.0  |
> |:-----------:|:-----:|:-----:|:-----:|:-----:|:-----:|:-----:|:-----:|:-----:|
> |  MegaDepth  | **0.106** | 0.107 | 0.107 | 0.107 | 0.108 | 0.108 | 0.109 | 0.109 |
> |   ScanNet   | **0.098** | 0.104 | 0.108 | 0.108 | 0.107 | 0.108 | 0.108 | 0.109 |
> |    Waymo    | **0.149** | 0.151 | 0.154 | 0.157 | 0.158 | 0.157 | 0.157 | 0.157 |
>
> Interestingly, we observe performance improvement when sampling near the border $(k=0.7)$, which partially supports our argument. For instance, the improvement in ScanNet is from $0.109$ to $0.098$.
>
> **R1, Q4: Minimal image content to predict the incidence field.**
>
> We believe that estimating the incidence field is easier on images with projective distortions. Following this reasoning, we consider object-centric images challenging, as the camera is too close to observe the projective distortion.
>
> Yet, a trade-off exists. Although estimating the incidence field in object-centric images is challenging, its accuracy has less impact to 3D sensing. In other words, an accurate 3D structure remains feasible despite noisy intrinsic parameters. A supporting fact is that many 3D object and 3D face modeling works use predefined intrinsic or weak projection to render their 3D model onto 2D images. We give two examples:
> - [img2pose: Face Alignment and Detection via 6DoF, Face Pose Estimation, CVPR'21]
> - [Fully understanding generic objects: Modeling, segmentation, and reconstruction, CVPR'21]
>
> **R1, Q5: Limitation in image hard to predict Incidence Rays.**
>
> We agree! An uncertainty measurement about the incidence field prediction will be beneficial!

---

> > ### Comment · Reviewer_TGwc · 2023-08-19
> > **Response**
> >
> > Thank you for your clarifications and further insights. After reading the other reviews and comments I will keep my initial positve rating.

---

### Author Rebuttal · Authors · 2023-08-10

**To all Reviewers:**
We value the reviewers' recognition of our method's novelty (R1, R3, R4, R5) and its strong performance (R1, R3, R4). We also thank reviewers R1 and R5 for recognizing the lucidity of our paper's explanations.

We present a method for calibrating $4$ DoF intrinsic of in-the-wild monocular images. Unlike other works, our method can be trained over non-perspective images. Beyond calibration, we showcase compelling downstream applications, including the detection of image cropping and uncalibrated two-view pose estimation.

**Convention:**
We refer to main and supplementary figures/tables/references as numbers (Fig. 1, Tab. 1, [1]). In contrast, we refer to rebuttal figures/tables/references as alphabets (Fig. A, Tab. A, [A]).

**One-Page PDF:**
We've appended extra figures and tables to the one-page PDF.

- Fig. $A$:
We demonstrate an interesting downstream application by enhancing the in-the-wild monocular 3D object detection work [Omni3D, CVPR'23] using our inferred intrinsic parameters.

- Tab. $A$:
	We additionally benchmark the baseline [26] after being trained with substantially more data. This checkpoint is released after the NeurIPS'23 submission deadline. Also, we provide the baseline results which employ the minimal solver outlined in Section $3.2$.

- Tab. $B$:
We provide interesting joint perspective-intrinsic calibration results. The method is elaborated in response to R3, Q $2\& 3$, point $2$.

---

### Decision · Program_Chairs · 2023-09-21

**Decision:**

Accept (poster)

**Comment:**

This paper introduces an intriguing approach for monocular 4DoF intrinsic camera parameters. Reviewers favor the proposed method and the demonstrated results. The authors provided comprehensive feedback regarding specific questions and additional demonstration requests. The authors also provided a supplement pdf file that shows interesting downstream applications. There were concerns from reviewer wjnN, but AC notes that the authors adequately defend it. After the rebuttal phase, most reviewers voted for the positive scores. AC recommends paper acceptance. It is strongly advised to apply the valuable discussions and additional materials in the revision.